# Autodesmotic reactions for general strain energy evaluation in polycyclic aromatic nanocarbons
Yang Wang [1,2] ✉

Strain energy fundamentally shapes the structure, stability, and reactivity of $\pi$-conjugated nanocarbons, making its accurate quantification essential for rational molecular design. However, existing approaches rely on arbitrary reference choices, overlook critical $\pi$-energy balance, or demand extensive computations, limiting their reliability and scope. Here we introduce autodesmotic reactions, a general and efficient framework that maps any strained $\pi$-conjugated nanocarbon onto an operationally defined single-molecule reference while preserving molecular topology and ensuring proper $\pi$-energy balance. This reference resides within a virtual chemical space constructed from physically motivated models trained on planar benzenoid hydrocarbons. Benchmarking across diverse carbon nanobelts confirms the accuracy and robustness of the method, and applications to circulenes, helicenes, bowl-shaped hydrocarbons, nanotubes, and fullerenes demonstrate its broad versatility and reveal insightful strain–structure–property relationships. By resolving the fundamental and computational limitations of established methods, autodesmotic reactions provide a rigorous, general, and highly efficient route to strain energy evaluation, requiring only a single quantum chemical calculation per molecule. As a conceptual advance linking topology, $\pi$-energy, and strain, this framework lays a foundation for accelerated design of strained aromatic nanocarbons and offers a platform readily extensible via emerging machine-learning strategies.

Molecular strain[1,2] is a fundamental factor that profoundly influences the structure, stability, properties, and reactivity of polycyclic aromatic nanocarbons. In helicenes[3], steric congestion twists the $\pi$-system into a helicoidal shape, imparting chirality and affecting racemization barriers. Fullerenes' stability largely depends on the distribution of strain over their carbon cages[4–7]. Curvature-induced strain in carbon nanotubes (CNTs) modulates their electronic characteristics[8], while macrocyclic strain in carbon nanobelts (CNBs)[9] enhances solubility[10] and facilitates functionalization[11,12]. Moreover, strain is a key determinant of reactivity and selectivity in aromatic nanocarbons[13], driving skeletal rearrangements[14] in polycyclic aromatic hydrocarbons (PAHs) and directing reactions to the most pyramidalized carbon sites in fullerenes[15] and bowl-shaped PAHs[16].

Quantifying strain energy (SE) can provide critical guidance for the rational design and synthesis of complex aromatic nanocarbons, where intrinsic strain often poses a formidable obstacle to successful preparation[9,17]. A striking example is the recent breakthrough synthesis of an armchair Möbius CNB[18]. Density functional theory (DFT) calculations[19] indicated that the SE enabling the previously achieved synthesis of a (6,6)

CNB[20] was about 40 kcal/mol. Guided by this benchmark, a sufficiently large (25,25) Möbius CNB with an SE of 49.4 kcal/mol was identified as a feasible target and successfully synthesized[18]. In contrast, attempts to prepare a smaller (15,15) Möbius CNB failed, as expected from its much higher SE (85.7 kcal/mol)[18].

However, defining and evaluating SE remains challenging, due to its non-observable nature[21] arising from multiple interdependent effects, including bond length variation, bond angle bending, torsional strain, nonbonded interactions between close atoms, $\pi$-bond deformation, and, albeit rarely, electrostatic strain[1,2]. SE is therefore commonly expressed as the energy difference relative to (real or hypothetical) "strain-free" references with idealized geometries and without nonbonded steric repulsion[22]. However, the choice of such reference structures is inherently arbitrary[23], which underlies the fundamental differences among existing SE evaluation approaches.

Compared to earlier group increment methods[24,25], reaction-based schemes[26] are widely used to estimate the SEs of aromatic nanocarbons via quantum chemical calculations. In isodesmic and (hyper)homodesmotic

[1]School of Chemistry and Chemical Engineering, Yangzhou University, Yangzhou, Jiangsu, China. [2]Jiangsu Provincial Key Laboratory of Green and Functional Materials and Environmental Chemistry, Yangzhou University, Yangzhou, Jiangsu, China. ✉e-mail: yangwang@yzu.edu.cn

reactions[26], reference molecules are constructed to match the target in both numbers and types of bonds and atoms, aiming to cancel all nonstrain contributions. Nonetheless, multiple valid reference sets exist, and this arbitrariness often produces ambiguous SE values. Biased reference selection may further unbalance other confounding effects, such as proto-branching interactions[23] or hyperconjugation[27], compromising SE accuracy.

A particularly critical and, surprisingly, long-overlooked issue for aromatic nanocarbons is the potential imbalance of $\pi$-energy (aromaticity) between the target molecule and its references. For instance, Fig. 1 shows three valid homodesmotic reactions for [5]helicene, yielding widely divergent SE values (4.7 to 81.0 kcal/mol) due to $\pi$-energy imbalance in all cases. Even the more stringent hyperhomodesmotic reactions may suffer from the same problem (see Supplementary Note 1).

To make matters worse, reactions designed to be ostensibly homodesmotic are generally not strictly homodesmotic[26] for polycyclic $\pi$-conjugated systems, because single and double carbon–carbon bonds are ill-defined in such delocalized bonding situations, which should be described by multiple resonance structures[28]. This fundamental flaw renders homodesmotic reactions intrinsically incapable of providing reliable SE evaluations for $\pi$-conjugated nanocarbons.

Only for specific systems can these limitations be partially circumvented by two recently proposed approaches[19,29], both of which exploit the macrocyclic structures of CNBs and related systems. The first, referred to as the "asymptotic model", assumes that the SE of a CNB with $n$ repeat units scales inversely with $n$[19] and estimates it by regressing DFT energies for a series of CNBs with varying $n$. This method may fail when other size-dependent effects (e.g., steric repulsion in helicene-containing CNBs[30]) are significant, requiring nontrivial corrections. The second approach, StrainViz[29], removes a segment from the target macrocycle, relaxes the resulting fragment to its unstrained geometry, and derives local SEs from energy–force relationships along the optimization trajectory. Accuracy is improved by averaging over fragments cut at different sites. StrainViz is unsuitable for macrocycles whose fragments relax into nonplanar structures, such as those containing helicene-like subunits. Both approaches are restricted to macrocyclic systems, and evaluating the SE of a single molecule demands substantial effort to construct and compute a large number of model structures.

For this reason, despite their inherent limitations and fundamental reliability concerns, homodesmotic reactions remain widely used as a general and convenient means of estimating SEs in strained aromatic nanocarbons. Even so, for complex $\pi$-conjugated molecules, designing workable homodesmotic reactions becomes highly challenging. For example, to estimate the SE of buckminsterfullerene $C_{60}$, a multi-step sequence of seven homodesmotic reactions was proposed, inspired by experimentally known dehydrogenation of a PAH molecule to $C_{60}$[31]. Consequently, SE

predictions based on homodesmotic reactions have been very rarely reported for fullerenes and related structures.

In this work, we introduce the concept of "autodesmotic reactions" as an efficient, robust and broadly applicable solution for SE evaluation in polycyclic aromatic nanocarbons. Unlike conventional methods that typically rely on multiple reference compounds, an autodesmotic reaction transforms the target molecule into a single-molecule reference. This hypothetical reference is systematically defined within a virtual strain-free chemical space, constructed by physically motivated model fitting to a large training set of planar benzenoid PAHs. A key feature of this approach is the preservation of molecular topology, ensuring ideal $\pi$-energy balance, a prerequisite for accurate SE assessment in $\pi$-conjugated systems. Our method delivers SE predictions for benchmark cases that closely match results from the asymptotic and StrainViz models, while requiring only a single quantum chemical calculation per molecule. It directly applies to nonmacrocyclic and challenging systems, for which the asymptotic, StrainViz, and other methods are not applicable. Applications to CNBs, circulenes, helicenes, bowl-shaped PAHs, CNTs, and fullerenes demonstrate the reliability, efficiency, and broad applicability of autodesmotic reactions, providing deeper insight into the strain–structure–property relationships of $\pi$-conjugated molecules and establishing a general framework for SE analysis in aromatic nanocarbons.

## Results
### The concept of autodesmotic reactions
In strained $\pi$-conjugated molecules, aside from bond length and bond angle distortions as well as nonbonded steric interactions, the twisting and/or bending of the $\pi$-system provides an indispensable contribution to the total strain[1,2]. We therefore assume that the SE in a PAH originates primarily from three sources:

(1) Out-of-plane strain—destabilization of $\pi$-conjugation caused by nonplanarity of the carbon framework, which reduces the overlap between $\pi$ atomic orbitals[1];
(2) In-plane strain—energy increase due to deviations in bond lengths and bond angles from their ideal values;
(3) H···H repulsion—steric interactions between hydrogen atoms in close spatial proximity (when present)[32].

Accordingly, to rigorously evaluate SE using a single-molecule reference, the idealized strain-free reference should meet the following conditions:

(1) Fully planar geometry;
(2) All carbon bond angles equal to 120°;
(3) All carbon–carbon bond lengths at their equilibrium values;
(4) Absence of close H···H contacts;
(5) Identical atomic connectivity to that of the strained target molecule.

Crucially, the last criterion's preservation of the carbon framework topology ensures that the reference retains essentially the same $\pi$-energy that the strained target molecule would possess if hypothetically relaxed to an ideal, unstrained state. This follows from Hückel molecular orbital (HMO) theory[33], in which the $\pi$-energy of an idealized conjugated carbon system depends solely on its atomic connectivity. As a result, any difference in $\pi$-energy between the target molecule and its reference originates effectively from geometric or steric distortions, rather than from variations in $\pi$-bonding patterns dictated by atomic connectivity.

Once such a hypothetical reference structure can be identified, we define an "autodesmotic reaction" as a unimolecular transformation that converts a given strained molecule into its corresponding strain-free reference isomer with identical atomic connectivity, while satisfying the five criteria outlined above. The operational definition of an autodesmotic reaction is schematically summarized in Fig. 2. The SE of the target molecule is then evaluated as the energy difference between strained structure and its reference (Fig. 2a, b). Unlike isodesmic and (hyper)homodesmotic reactions, autodesmotic reactions ideally balance the energies of $\pi$-conjugated

**Fig. 1 | Homodesmotic reactions for evaluating the SE of [5]helicene using different reference sets. a** benzene and naphthalene, (**b**) pentacene, (**c**) dibenzo[bc,kl] coronene and coronene. The change in $\pi$-energy ($\Delta E_\pi$) for each reaction is evaluated using the predictive model described in the Methods Section.

In the figure:
(a) [structure] $+ 3$ [benzene] $\xrightarrow[\Delta E_\pi = +10.8]{SE = 10.7}$ $4$ [naphthalene]

(b) [structure] $\xrightarrow[\Delta E_\pi = +21.7]{SE = 4.7}$ [pentacene]  Energies in kcal/mol

(c) [structure] $+ 3$ [dibenzo structure] $\xrightarrow[\Delta E_\pi = -59.2]{SE = 81.0}$ $\frac{14}{3}$ [coronene]

**Fig. 2 | Schematic overview of the autodesmotic reaction concept. a** A strained molecule is transformed into (**b**) its corresponding strain-free reference structure with identical atomic connectivity. **c** A virtual strain-free chemical space is constructed by model fitting to DFT-computed energies and geometries of (**d**) a large set of planar benzenoid PAHs. Mapping the target molecule into this space while preserving atomic connectivity yields the energy and geometry of its reference via model-based interpolation.

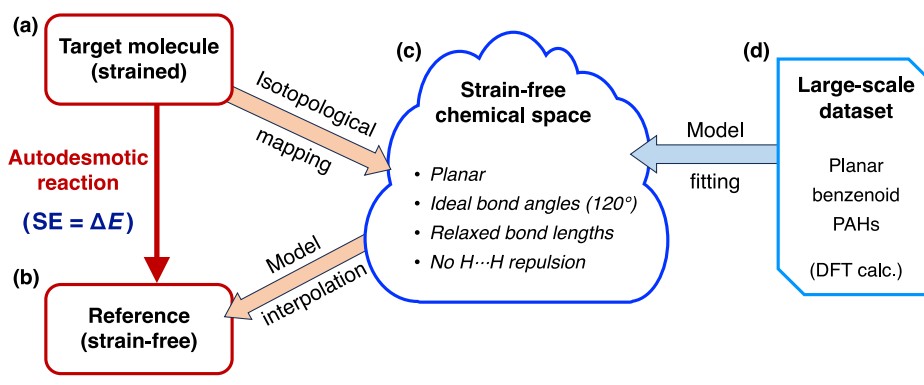

**Fig. 3 | Benchmarking of ten C$_{48}$H$_{24}$ carbon nanobelts. a** Ten C$_{48}$H$_{24}$ CNBs used for benchmarking, including the experimentally synthesized CNB (6,6)-16[36,37]. In each CNB, the repeat unit used in the asymptotic model is highlighted in blue. Hydrogens are omitted for clarity. **b** Comparison of SEs obtained from the autodesmotic (blue), StrainViz[29] (orange), and asymptotic[19] (green) models for these CNBs.

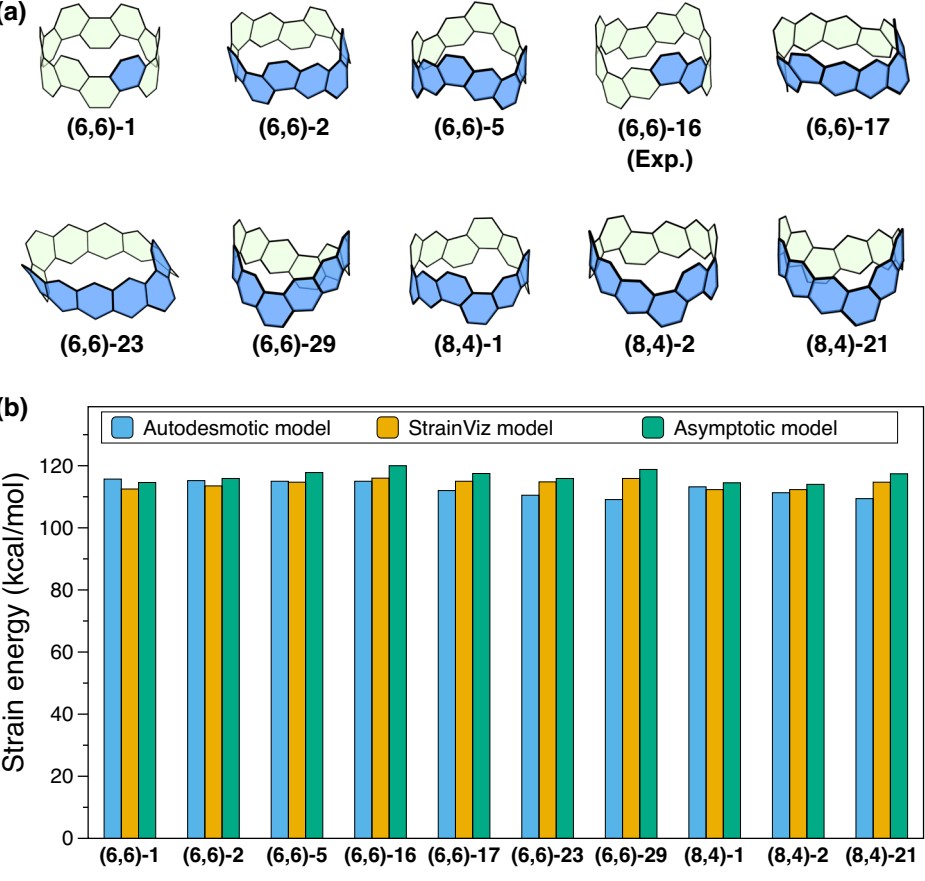

systems between the target and reference molecules for accurate SE evaluation. Moreover, their unimolecular nature effectively eliminates the basis set superposition error[34] commonly present in (hyper)homodesmotic reactions, which arises from size inconsistencies between the target and reference molecules.

The key challenge lies in identifying this autodesmotic reference structure and estimating its associated energy, as both are typically hypothetical. To tackle this, our strategy is to interpolate the equilibrium structure and energy of the reference within a chemical space constructed or learned from a large collection of well-characterized molecules. Specifically, we construct the strain-free chemical space (Fig. 2c) by model fitting to DFT-computed energies and geometries of a broad and diverse dataset of benzenoid (composed exclusively of hexagonal rings) PAHs (Fig. 2d). All molecules in this dataset possess planar equilibrium geometries and closed-shell electronic ground states. Two predictive models are then developed using chemically meaningful descriptors and multiple regression: one for

molecular energies and one for equilibrium geometries. Together, these models define the virtual chemical space within which the desired autodesmotic reference can be located by mapping the target molecule into the space while preserving atomic connectivity. All methodological details are described in the Methods section.

**Validation and efficiency comparison of the autodesmotic model against established SE methods**

We first benchmarked our autodesmotic model against the well-established StrainViz[29] and asymptotic[19] models within the C$_{48}$H$_{24}$ isomeric space. We selected ten C$_{48}$H$_{24}$ CNBs (see Fig. 3a for structures and ref. 35 for nomenclature), ensuring that both the StrainViz and asymptotic models are applicable. Among all enumerated C$_{48}$H$_{24}$ CNBs, these molecules display a well-defined quadratic relationship between the total energy and the inverse square of the belt size, fulfilling the prerequisite for the validity of the asymptotic model[19].

**Table 1 | CPU times (hours) and number of DFT calculations for autodesmotic, StrainViz, and asymptotic methods evaluating SEs of benchmark $C_{48}H_{24}$ CNBs**

| CNB | Autodesmotic Time ($N_{calc}$)[a] | StrainViz Time ($N_{calc}$)[b] | Asymptotic Time ($N_{calc}$)[c] |
|---|---|---|---|
| (6,6)-1 | 0.98 (1) | 45.20 (25) | 2171.30 (55) |
| (6,6)-2 | 4.70 (1) | 93.40 (25) | 883.12 (10) |
| (6,6)-5 | 2.24 (1) | 40.29 (25) | 844.58 (19) |
| (6,6)-16 | 1.74 (1) | 44.87 (25) | 1120.84 (28) |
| (6,6)-17 | 2.47 (1) | 41.12 (25) | 507.72 (10) |
| (6,6)-23 | 2.48 (1) | 41.37 (25) | 626.94 (10) |
| (6,6)-29 | 11.44 (1) | 56.91 (25) | 1019.42 (19) |
| (8,4)-1 | 1.84 (1) | 45.38 (25) | 1387.13 (38) |
| (8,4)-2 | 2.31 (1) | 44.01 (25) | 837.78 (19) |
| (8,4)-21 | 2.07 (1) | 38.61 (25) | 542.42 (19) |

[a]A B3LYP/6-311G*//B3LYP/6-31G* calculation; numbers in parentheses (italic) indicate $N_{calc}$, the number of DFT calculations.
[b]B3LYP/6-31G* geometry optimizations; numbers in parentheses (italic) indicate $N_{calc}$.
[c]B3LYP/6-31G* geometry optimizations for CNBs with up to 60 benzenoid rings; numbers in parentheses (italic) indicate $N_{calc}$.

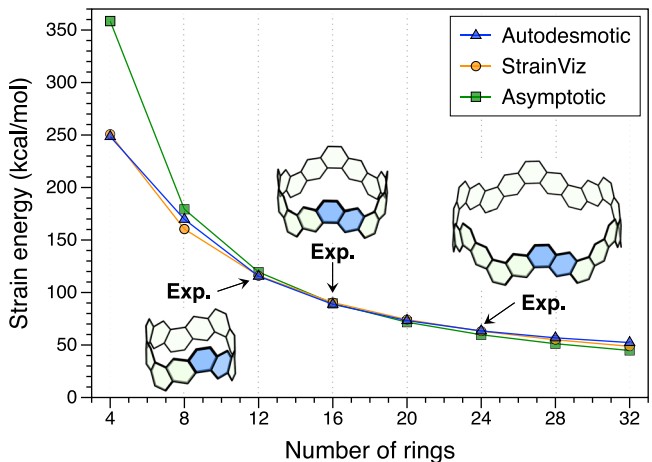

**Fig. 4 | SE predictions for armchair (n, n) CNBs of varying sizes (n = 2, 4, …, 16).** Results from the general autodesmotic model are (blue triangles) compared with those from the StrainViz[29] (orange circles) and asymptotic[19] (green squares) models. The insets show the structures of the three experimentally synthesized CNBs: (6,6)-16[36,37], (8,8)-201[38], and (12,12)-12235[37].

As shown in Fig. 3b, the SEs predicted by the autodesmotic model are in good congruence with those obtained from the StrainViz and asymptotic models. The mean absolute percentage differences relative to the autodesmotic predictions are about 3%. Notably, for the experimentally synthesized CNB (6,6)-16[36,37], the SE obtained from the autodesmotic model is 115.0 kcal/mol, essentially identical to the StrainViz prediction (116.0 kcal/mol) and reasonably consistent with the asymptotic model result (120.0 kcal/mol).

Although all three methods yield similar SE predictions for the benchmark CNBs, our autodesmotic approach offers substantially higher computational efficiency than the StrainViz and asymptotic methods. As shown in Table 1, it requires only a single DFT calculation per molecule, with total CPU times of typically just a few hours for each CNB. By contrast, StrainViz involves multiple geometry optimizations and generally consumes an order of magnitude more time, while the asymptotic method is even more computationally demanding, requiring two to three orders of magnitude longer. This dramatic reduction in both the number of calculations and overall computational cost highlights the efficiency of the

autodesmotic method, making it particularly advantageous for studying large and complex aromatic nanocarbons.

We further validated the reliability of the autodesmotic method by comparing its SE predictions with StrainViz results for diverse $C_{48}H_{24}$ CNBs, for which the asymptotic model is inapplicable. These CNBs encompass a variety of geometric shapes, conformations, and topological types, including tubular, conical, quasiplanar, and Möbius nanobelts. As shown in Supplementary Fig. S2, the autodesmotic SE predictions are generally consistent with StrainViz and capture chemically sensible trends.

**Armchair CNB series with experimentally synthesized structures**

To assess the performance of the general autodesmotic model for molecules of varying sizes and compositions, we applied it to a series of armchair (n, n) CNBs (n = 2, 4, …, 16) sharing a common repeat unit (highlighted in blue in the inset of Fig. 4). This series includes several experimentally synthesized CNBs, namely, (6,6)-16[36,37], (8,8)-201[38], and (12,12)-12235[37], which serve as practical benchmarks for assessing the accuracy of SE models.

For the experimentally synthesized [12]CNB (6,6)-16 ($C_{48}H_{24}$)[36,37], the general autodesmotic model and the $C_{48}H_{24}$-specific autodesmotic model yield closely matching SEs (114.4 and 115.0 kcal/mol, respectively), despite being trained on entirely distinct datasets. This agreement demonstrates the reliable generalization capability of the general autodesmotic model.

Figure 4 compares the SEs predicted from the general autodesmotic, StrainViz, and asymptotic models as a function of the number of rings in the CNB. The three models produce similar predictions for CNBs containing 12 or more rings. For [8]CNB, noticeable discrepancies arise among the models, with the autodesmotic prediction (blue triangle) lying between the StrainViz (orange circle) and asymptotic (green square) predictions. For the smallest system, [4]CNB, the autodesmotic and StrainViz models remain in close agreement (243.7 vs. 250.8 kcal/mol), whereas the asymptotic model significantly overestimates the SE (358.5 kcal/mol). This large deviation is due to the fact that the SE predicted by the asymptotic method is an extrapolated reference value, obtained without optimizing [4]CNB (see Supplementary Note 3 for the detailed regression analysis). It is assumed to represent the SE of (2,2) CNB while preserving the aromaticity of the benzene rings, rather than adopting the stable quinoidal structure.

**Structurally unconventional CNBs**

As the first test case, the nonstandard[35] CNB (9,0,3)-1 serves as a representative challenging example, featuring helicene substructures that induce significant structural distortion due to close H⋯H contacts (highlighted by pink dotted lines in Fig. 5a). This H⋯H repulsion varies nonlinearly with CNB size, breaking the inverse relationship[19] between SE and belt size and rendering the asymptotic model inapplicable. To address this, a modified asymptotic approach combined with conventional homodesmotic reactions was proposed to estimate the SEs of these helicene-containing CNBs[30]. Nevertheless, designing appropriate homodesmotic reactions is a nontrivial task, entailing the laborious construction of reference structures, while the balance of π-conjugation energy remains unclear. For CNB (9,0,3)-1, our autodesmotic model predicts an SE of 157.1 kcal/mol, which is considerably higher than the SEs of helicene-free CNBs (about 110–120 kcal/mol; see Fig. 3 and Supplementary Fig. S2), implying the additional strain imposed by H⋯H repulsion in helicene-containing CNBs. By comparison, the combined asymptotic-homodesmotic method estimates the SE at 165.1 kcal/mol[30], showing satisfactory consistency with the autodesmotic result.

As the next examples, [n]circulenes are a distinctive family of macrocyclic arenes featuring an n-membered central ring fused with n benzenoid rings around the periphery[39]. Circulenes larger than the planar, strain-free [6] circulene (coronene) adopt strained, saddle-shaped structures due to crowding among the peripheral rings. Consequently, [7]circulene (Fig. 5b) is the only larger circulene that has been successfully synthesized to date[40]. Although a derivative of [8]circulene has been isolated[41], the synthesis of its pristine form remains a significant challenge. The SEs of [7]- and [8]circulenes, determined via autodesmotic reactions, are 55.5 and 103.1 kcal/mol, respectively, which helps rationalize the observed difference in their synthetic

**Fig. 5 | SE predictions for various types of π-conjugated hydrocarbons. a** Helicene-containing CNB (9,0,3)-1[30], with close H ⋯ H contacts highlighted in pink. **b** Circulenes[39] and septulene[42]. **c** Helicenes[43]. **d** Corannulene[45] and carboncone[46]. The SEs predicted in this study and reported in previous works[30,39,42,45,46] are shown in blue and brown, respectively. Green numbers indicate the experimentally measured racemization barriers of helicenes at around room temperature[43].

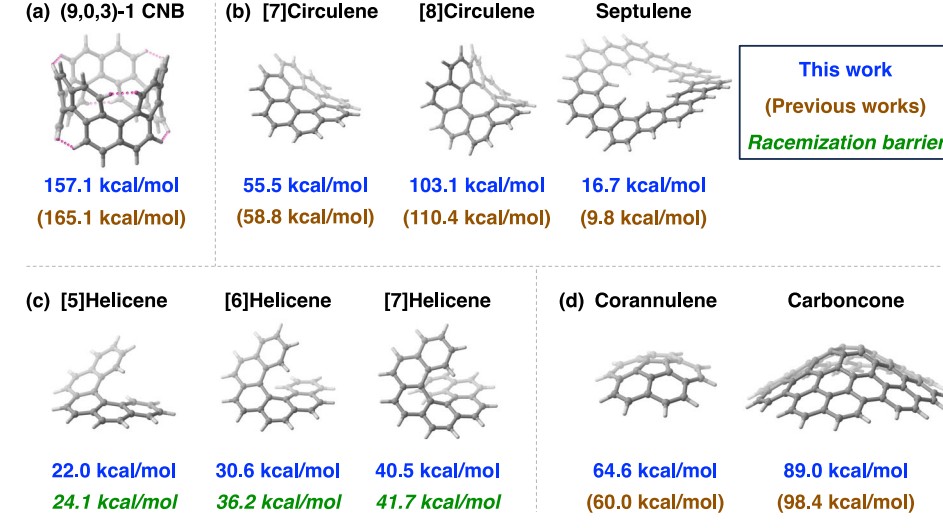

accessibility. These results agree with estimates from a prior study (58.8 and 110.4 kcal/mol) based on a group increment method utilizing the $C_4H_2$ unit in circulenes as the group equivalent[39]. Incidentally, our StrainViz calculations for these circulenes failed because the relaxed fragment obtained after removing a $C_{10}H_6$ unit corresponds to a strained [4]- or [5]helicene in the cases of [7]- and [8]circulene, respectively. Although StrainViz calculations for [7]circulene can proceed by removing a larger $C_{14}H_8$ substructure during fragment construction, this compromise likely introduces greater errors in SE prediction due to substantial disruption of the macrocyclic π-conjugation.

Septulene[42], the heptagonal homologue of kekulene, is another example of experimentally synthesized saddle-shaped CNBs. Compared with circulenes, the 14 benzenoid rings in septulene annulate around the belt periphery in a much less congested manner (Fig. 5b). As a result, it exhibits a considerably lower SE of 16.7 kcal/mol, as predicted by the autodesmotic model. In the original synthesis report[42], the SE of septulene was estimated to be only 9.8 kcal/mol based on its homodesmotic transformation to kekulene. However, considering that kekulene itself has an SE of 7.0 kcal/mol (Supplementary Fig. S2a), this underestimate should be corrected. Using the homodesmotic reaction 6 septulene → 7 kekulene, in which the π-energy is well balanced (see Supplementary Note 4), the corrected SE is obtained by adding the SE of kekulene scaled by stoichiometry: 9.8 + 7.0 × (7/6) = 18.0 kcal/mol, bringing it into agreement with the autodesmotic prediction.

### Helicenes and bowl-shaped PAHs

Helicenes[3] are an important class of PAHs, characterized by a helical skeleton of spirally fused benzenoid rings (Fig. 5c). Their unique molecular structure, inherent chirality, and extended π-conjugation make them attractive for diverse applications in asymmetric chemistry, chiroptics, and materials science[3]. Owing to their nonmacrocyclic structures, neither StrainViz nor the asymptotic model is applicable for evaluating the SEs of helicenes. Using autodesmotic reactions, the SEs of [5]-, [6]-, and [7]helicenes are predicted to be 22.0, 30.6, and 40.5 kcal/mol, respectively. Interestingly, both the magnitudes and trend of these SE values closely match their experimental barriers of racemization[43] (green numbers in Fig. 5c). Although experimental barriers for higher helicenes with eight or more rings have also been measured, their racemization process follows a complex multi-step mechanism mediated by multiple intermediates, as revealed by latest computational study[44]. Therefore, these higher helicenes are not discussed here.

Bowl-shaped PAHs[16] constitute another notable category of strained nanocarbons, often viewed as segments of fullerenes or as end caps of CNTs. One of the smallest members, corannulene (Fig. 5d), exhibits a relatively high SE of 64.6 kcal/mol, as determined by our autodesmotic calculation. A previous study[45], based on the homodesmotic reaction using coronene as a reference, reported an SE of 60.0 kcal/mol, in reasonable agreement with our

result. Notably, in that homodesmotic reaction the π-energy is approximately balanced (within 4 kcal/mol), as estimated from our energy prediction model (see Supplementary Note 4). In addition, our StrainViz calculations gave an SE of only 45.7 kcal/mol for corannulene. The underestimation likely arises from an overapproximation in StrainViz's fragmentation scheme, which omits a substantial portion ($C_{10}H_6$, nearly half of the molecule) to define each fragment, leading to the loss of important structural information from the intact molecule[29].

As a π-extended analogue of corannulene, carboncone and its derivatives have been synthesized via bottom-up routes[46], with the carbon framework unambiguously identified by X-ray crystallography (Fig. 5d). Using our autodesmotic model, we predict an SE of 89.0 kcal/mol for carboncone, substantially higher than that of corannulene. A previous calculation[46] estimated the SE of penta-mesityl substituted carboncone to be 98.4 kcal/mol, reasonably agreeing with the magnitude of our prediction for the unsubstituted skeleton. In that calculation, the SE was derived from a homodesmotic reaction using a large hexa-mesityl substituted benzenoid PAH as a strain-free reference, although the balance of π-energy was not assessed.

### Carbon nanotubes

CNTs are intriguing one-dimensional nanocarbons with fully delocalized π-electron systems. Owing to the intrinsic strain imposed by their seamless cylindrical structure, SE plays a fundamental role in determining their structural and thermal stability[47,48], growth mechanisms[49], electronic properties[8], as well as their mechanical behavior and chemical reactivity[50]. However, quantum chemical evaluation of SEs for finite-length CNTs[51] remains computationally intensive. While a CNT can be regarded as an axially extended analogue of a CNB, and its SE can, in principle, be estimated using the asymptotic and StrainViz models, such calculations are complicated by the structural construction procedure and become increasingly computationally demanding as the CNT length grows. In contrast, the autodesmotic methodology circumvents these difficulties by relying on a single DFT calculation for the target CNT, followed by straightforward application of our predictive models.

Figure 6a presents the SEs of a series of single-walled armchair CNTs, $(n, n)$, with varying lengths and radii ($n = 5$–8), as evaluated using the autodesmotic model. For each fixed radius (i.e., same $n$), the SE increases almost linearly with the increasing CNT length ($R^2 > 0.999$), indicating a nearly constant SE per carbon atom. Comparison of the regression lines in Fig. 6a shows that CNTs with larger radii exhibit a lower SE per carbon atom, as anticipated from their reduced curvature. Figure 6b reveals an almost perfect linear correlation ($R^2 = 0.99996$) between the SE per atom, averaged over all CNTs of the same radius, and the inverse square of the CNT radius ($1/r^2$). This relationship can be naturally derived from continuum elasticity

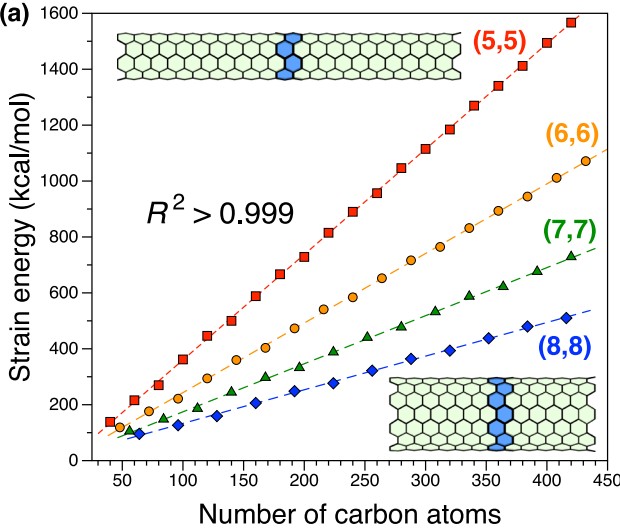

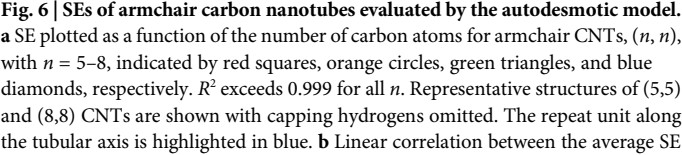

**Fig. 6 | SEs of armchair carbon nanotubes evaluated by the autodesmotic model.** **a** SE plotted as a function of the number of carbon atoms for armchair CNTs, $(n, n)$, with $n$ = 5–8, indicated by red squares, orange circles, green triangles, and blue diamonds, respectively. $R^2$ exceeds 0.999 for all $n$. Representative structures of (5,5) and (8,8) CNTs are shown with capping hydrogens omitted. The repeat unit along the tubular axis is highlighted in blue. **b** Linear correlation between the average SE

per carbon atom, $\varepsilon$, and the inverse square of average radius, $r$, for armchair CNTs (dark brown circles and line). Results for various chiral CNTs are shown as empty blue symbols: a square for (8,5), a diamond for (7,4), an upward-pointing triangle for (12,6), and a downward-pointing triangle for (6,5). All CNTs considered here have been experimentally produced and identified[51].

models[52]. The coefficient of proportionality obtained from our autodesmotic results is 42.36 kcal/(mol · Å²) per atom, in good agreement with values from previous first-principles simulations under periodic boundary conditions: 46.6[53] and 49.3 kcal/(mol · Å²) per atom[54]. This proportionality constant, closely related to the elastic properties of CNTs, allows derivation of the corresponding Young's modulus once a wall thickness is assumed[53,54].

In addition, the SEs of various chiral CNTs with distinct radii were readily computed using autodesmotic reactions. As shown by the empty blue symbols in Fig. 6b, the SE per carbon atom for these chiral CNTs closely follows the same $1/r^2$ relationship observed for armchair CNTs, indicating that the inverse-square law holds broadly, with a proportionality constant essentially independent of CNT chirality. This finding is further corroborated by objective molecular dynamics results reported previously[55].

### Fullerenes

SE evaluation has been a long-standing focus of fullerene research, deepening understanding of their structural stability[4–6], aromaticity[31,56], formation mechanisms[57], and chemical functionalization[58,59]. Because strain is intimately linked to aromaticity arising from full $\pi$-conjugation over the spherical surface of the carbon cage[31], accurately isolating and quantifying SE in fullerenes remains challenging.

Indeed, even for the archetypal buckminsterfullerene $C_{60}$, previous attempts have reported widely varying SE estimates. Cyrański et al.[56] constructed a model for C–C bond energies in fullerenes by fitting a bond-energy–bond-length relationship to atomization energies[60], and fitted similarly a reference model based on planar PAHs containing the same ring motifs as the fullerene. The SE was then quantified as the difference between the sums of C–C bond energies predicted by the fullerene and reference models. For $C_{60}$, this approach yielded an SE of 93.6 kcal/mol[56], much smaller than the 484 kcal/mol estimated from $\pi$-orbital axis vector (POAV) analysis[6]. The discrepancy stems mainly from the fact that the reference PAHs in the bond energy approach[56] already bear substantial strain induced by their pentagonal rings, despite being planar. A more recent estimate was provided by Suresh et al.[31] through a thoughtfully designed scheme involving seven homodesmotic reactions based on the experimentally known transformation of a $C_{60}H_{30}$ PAH into the $C_{60}$ fullerene. Their calculated SE of 381.4 kcal/mol differs markedly from the earlier bond energy[56] and POAV[6] estimates. Employing the autodesmotic model, we obtained an SE of 395.8 kcal/mol for $C_{60}$, which most strongly supports Suresh et al.'s prediction[31].

We next selected all 1259 fullerene isomers of $C_{60}$ with closed-shell ground states and computed their SEs using the autodesmotic model. Clearly, the experiment-inspired homodesmotic reaction scheme[31] designed specifically for buckminsterfullerene $C_{60}$ is not applicable to other cage isomers. As shown in Fig. 7, the SE of $C_{60}$ isomers correlates well ($R^2 = 0.867$) with cage sphericity, as measured by a metric related to the volume-to-area ratio[61], indicating that cages with more uniformly distributed curvature exhibit lower strain. This observation is in line with previous findings for neutral pristine fullerene isomers[7]. Furthermore, the SE of $C_{60}$ isomers generally follows both the isolated pentagon rule[4] and the pentagon adjacency penalty rule[5]: those having a greater number of adjacent pentagon pairs tend to have a higher SE, as indicated by the color scale in Fig. 7.

### Discussion

As demonstrated, autodesmotic reactions provide a powerful and general conceptual framework for evaluating the SE of polycyclic aromatic nanocarbons. In this approach, the SE is unambiguously defined as the energy difference between a given molecule and its hypothetical reference, which preserves the same atomic connectivity while adopting unstrained geometries free of steric hindrance. This single-molecule reference is located within a virtual strain-free chemical space, constructed from chemically intuitive models trained on a large set of planar benzenoid PAHs. By preserving molecular topology, autodesmotic reactions not only guarantee a well-defined reference but also ensure proper $\pi$-energy balance, addressing two critical factors often overlooked in SE analysis of conjugated systems. In essence, the autodesmotic reference represents the hypothetical unstrained counterpart of the curved molecule, carrying the ideal $\pi$-energy that the curved molecule would possess if it could be perfectly planarized. Although such a strain-free reference may not exist as a real molecule in three-dimensional space, it is operationally and reproducibly defined within the higher-dimensional chemical space, allowing that SE values are directly comparable across diverse curved molecules without reliance on arbitrarily chosen external references.

It is important to emphasize that SE is not a quantum-mechanical observable but an operationally defined quantity dependent on the chosen unstrained reference. Consequently, every method for estimating SE entails some degree of arbitrariness. In the autodesmotic approach, arbitrariness arises from the selection of training molecules used to construct the chemical space and from the analytical forms of the model equations. The aim

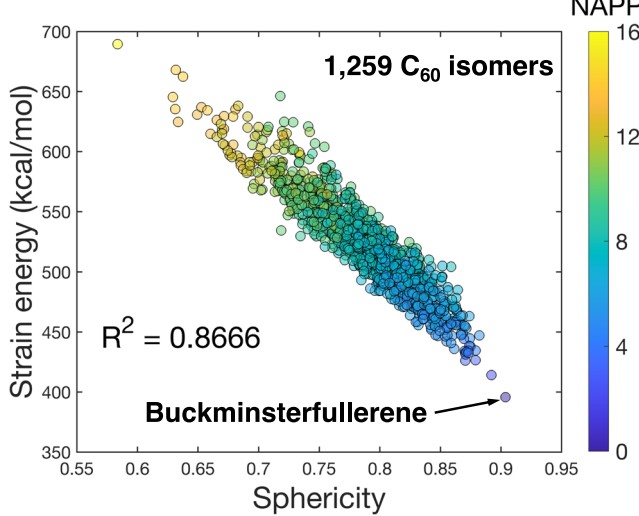

**Fig. 7 | SEs for $C_{60}$ fullerene isomers predicted by the autodesmotic model.** SE is plotted against cage sphericity[61] for all possible closed-shell fullerene isomers of $C_{60}$. Colors scale with the number of adjacent pentagon pairs (NAPP) of each cage.

of the present work is not to define a unique "strain energy operator", but to minimize ambiguity in a physically transparent and computationally efficient manner. By enforcing isotopological mapping, which preserves the atomic connectivity of the target molecule in its reference, autodesmotic reactions mitigate a key source of arbitrariness in $\pi$-conjugated systems, namely, the imbalance of $\pi$-energy between target and reference. Moreover, the approximations in the present implementation can be systematically refined through re-parameterization or by adopting advanced machine-learning schemes, thereby reducing arbitrariness and improving predictive accuracy.

Benchmarking against the asymptotic and StrainViz models across a diverse range of CNBs validates the autodesmotic approach as both accurate and robust. Its applicability extends to more challenging systems, including helicene-containing CNBs, saddle-shaped CNBs, helicenes, bowl-shaped PAHs, CNTs of varying radii and lengths, and numerous $C_{60}$ fullerene isomers. Remarkably, despite their substantial inherent strain, most of these molecules have been successfully synthesized and characterized, marking major milestones in synthetic chemistry. From a theoretical standpoint, however, they pose particular challenges for SE evaluation, and for most of them it is difficult or even impossible to apply established SE models. The predicted SEs align with established chemical knowledge and provide valuable insights into strain–structure–property relationships. A major advantage of the autodesmotic approach is that it requires only a single computation per target molecule, unlike other SE methods necessitating additional calculations on multiple reference or auxiliary structures. Consequently, the autodesmotic method is orders of magnitude more computationally efficient than the StrainViz and asymptotic methods, enabling rapid SE analysis in large, complex $\pi$-conjugated nanocarbons. We note that homodesmotic-like approaches may also achieve computational savings when pre-computed reference compounds are reused across a series of structurally related target molecules. However, to our knowledge, no rigorous and systematic framework exists that enables such reuse across diverse $\pi$-conjugated nanocarbons while maintaining consistency in reference selection. The autodesmotic framework provides such a systematic and generalizable route, formalizing procedures that are otherwise applied in an ad hoc manner. Building this virtual reference space requires a one-time computational investment, analogous to model training in machine learning, after which each new molecule in the same chemical family requires only a single DFT calculation for its SE evaluation.

The accuracy and generalizability of the present approach can be further enhanced by refining the energy- and geometry-prediction models

used to construct the strain-free reference space. Like all practical models, the autodesmotic framework is not free from limitations and will benefit from continued refinement. For instance, both total energy and equilibrium geometry predictions could be substantially improved by employing advanced modeling approaches beyond the physically motivated regression method implemented here. Among these, machine-learning techniques are particularly promising in light of the recent surge in artificial intelligence applications.

The current autodesmotic method has several limitations. First, its implementation is restricted to fully conjugated hydrocarbons and carbon allotropes containing only sp$^2$ carbons. Extending the framework to heteroatom-doped or functionalized systems will require redefining the reference chemical space to include representative heteroatomic and mixed-hybridization motifs, or alternatively, training fully data-driven models that learn directly from diverse molecular datasets. Second, the method evaluates SE only relative to six-membered benzenoid units, treating all deviations from hexagonal topology as sources of strain. Expanding the model to include non-benzenoid PAHs containing four- and five-membered rings would enable autodesmotic references for molecules built from such units, allowing quantification of SE associated with framework bending and macrocycle closure in non-benzenoid systems. This extension provides more directly useful information for synthetic chemists. Finally, while the current framework quantifies total SE globally for the entire molecule, it does not yet allow decomposition into specific bonds, angles, or regions, for which StrainViz has proven effective. This limitation results from the delocalized nature of $\pi$ electrons, which prevents rigorous bond-wise partitioning of $\pi$-energy. Developing strategies for spatially resolved SE decomposition is therefore an important methodological challenge and an intriguing direction for future work.

In summary, autodesmotic reactions lay a general, conceptually transparent, computationally efficient, and extendable framework for SE evaluation in $\pi$-conjugated systems, guiding the understanding, rational design, synthesis, and application of strained nanocarbons.

## Methods
### Energy prediction model in isomeric space
We begin with the energy prediction model for PAHs in isomeric space. As a representative demonstration, we first consider all candidate $C_{48}H_{24}$ PAHs and develop a theoretical model describing their total energies. As discussed in the Results section, all candidate molecules in the reference space must possess planar, benzenoid structures with a closed-shell ground state. Among the possible 114,326 $C_{48}H_{24}$ benzenoid isomers enumerated using the boundary-edges code algorithm[62], we identified 1516 legitimate isomers that satisfy these criteria for reference molecules (see Supplementary Note 5 for details).

We assume that the absolute total energy, $E_{tot}$, obtained from DFT calculations for any reference PAH molecule can be decomposed as

$$E_{tot} = E_\pi + E_{C-C} + E_{H\cdots H} + E_0 \qquad (1)$$

where $E_\pi$ is the energy of the $\pi$-conjugated system; $E_{C-C}$ accounts for contributions from the $\sigma$ bonds within the carbon framework; $E_{H\cdots H}$ represents the steric repulsion energy between close H$\cdots$H contacts[32]; and the constant $E_0$ includes all remaining contributions, such as those from C–H bonds and core electrons. The energy contributions from C–H bonds can be reasonably treated as constant[63], as their bond lengths show little variation throughout datasets (1.0860 ± 0.0018 Å; see Supplementary Fig. S5a,b). Note that "C–C bonds" does not necessarily denote single bonds, but refers generically to all carbon–carbon bonds regardless of bond order or type. Furthermore, since all reference PAHs are benzenoid structures composed exclusively of hexagonal rings, the C–C–C bond angles fluctuate around the ideal value of 120° (e.g., 120.37° ± 1.46° for the $C_{48}H_{24}$ dataset; see Supplementary Fig. S6). Such minor distortions in bond angles should have negligible influence on the total energy[2] and are therefore not accounted for in Eq. (1).

The first three terms on the right-hand side of Eq. (1) are evaluated at the equilibrium geometry of the given PAH molecule. Within the framework of simple HMO theory, the $\pi$-energy is computed as $E_\pi = \alpha n + 2\beta \sum_{i=1}^{n/2} \chi_i$, where $\alpha$ and $\beta$ are Coulomb and resonance integrals, respectively; $n = 48$ is the total number of carbon atoms; $\chi_i$ is the $i$th eigenvalue (in descending order) of the adjacency matrix of the molecular graph; and the summation runs over all doubly occupied Hückel orbitals.

The $E_{C-C}$ term is assumed to be a sum of independent energy contributions from individual C–C bonds[63], each expressed as a function of bond length: $E_{C-C} = \gamma \sum_i^{C-C} (d_i/d_0 - 1)^\lambda$, where $d_i$ is the $i$th C–C bond length; $d_0$, $\gamma$ and $\lambda$ are constant parameters determined through model fitting. The monotonic behavior of this function is supported by previous studies based on experimental and computational data[56,64], confirming its validity at least within our examined range of bond lengths from 1.35 to 1.48 Å. The total H⋯H repulsion energy is evaluated as $E_{H\cdots H} = \eta\, n_{H\cdots H}$, where $\eta$ denotes an empirical parameter representing the average repulsion energy per close H⋯H contact, and $n_{H\cdots H}$ is the total number of such contacts. Two hydrogen atoms are considered close and contribute to $n_{H\cdots H}$ if their distance is less than 2.05 Å, based on the distribution of H⋯H distances across all reference molecules (see Supplementary Fig. S5c,d). This coarse-grained treatment of overall H⋯H steric repulsion keeps the model simple yet effective, numerically stable, and broadly transferable. Tests show that incorporating explicit pairwise H⋯H repulsion using Lennard–Jones or Buckingham potentials yields no improvement, or only marginal improvement, over the coarse-grained model. Furthermore, such distance-dependent potentials introduce additional fitting parameters, which reduce numerical stability in model optimization and compromise model generalizability.

Under these assumptions, the total energy of a reference PAH molecule is given by

$$E_{tot} = 2\beta \sum_{i=1}^{n/2} \chi_i + \gamma \sum_i^{C-C} (d_i/d_0 - 1)^\lambda + \eta\, n_{H\cdots H} + \kappa \quad (2)$$

where $\kappa = \alpha n + E_0$ is a constant for all reference PAH isomers.

## Energy prediction model in heterogeneous chemical space

More generally, for PAHs with different sizes and compositions, the energy prediction model must be extended using an alternative strategy, because directly fitting the absolute total energies with Eq. (1) poses two major difficulties. First, the $E_0$ term in Eq. (1) is no longer a constant term but varies with chemical composition. Second, total energies span almost two orders of magnitude in our dataset due to differences in molecular size, whereas the desired fitting accuracy is typically within a few kcal/mol. These issues hinder the attainment of sufficient accuracy in model fitting. To overcome them, we fit the energy change $\Delta E_{ref}$ associated with a reference reaction, rather than the absolute total energies. The reference reaction is defined as

$$\frac{4m - n}{18} C_6H_6 + \frac{n - m}{72} C_{96}H_{24} \rightarrow C_nH_m + \Delta E_{ref} \quad (3)$$

where the reference molecules $C_6H_6$ and $C_{96}H_{24}$ represent benzene and circumcircumcoronene, respectively. Using the energy decomposition expressed in Eq. (1), the reference reaction energy, $\Delta E_{ref}$, is computed as

$$\Delta E_{ref} = (E_\pi + E_{C-C} + E_{H\cdots H} + E_0) - \frac{4m-n}{18}(E_\pi^* + E_{C-C}^* + E_0^*) \\ - \frac{n-m}{72}(E_\pi^{**} + E_{C-C}^{**} + E_0^{**}) \quad (4)$$

where superscripts * and ** refer to benzene and circumcircumcoronene, respectively. Note that H⋯H repulsion is absent in both benzene and circumcircumcoronene as neither molecule possesses H⋯H close contacts.

Applying the expressions for $E_\pi$, $E_{C-C}$, and $E_{H\cdots H}$ (see Eq. (2)), the reference reaction energy can be modeled using the following equation (see

Supplementary Note 7 for derivation):

$$\Delta E_{ref} = 2\beta \sum_{i=1}^{n/2} \chi_i + \gamma \sum_i^{C-C} (d_i/d_0 - 1)^\lambda + \eta\, n_{H\cdots H} + \nu\, n + \mu\, m + \epsilon \quad (5)$$

where parameters $\beta$, $\gamma$, $\lambda$, $d_0$, $\eta$, $\nu$, $\mu$, and $\epsilon$ are determined by least-squares fitting. Based on the predicted value of $\Delta E_{ref}$, the total energy of a given $C_nH_m$ molecule is given by

$$E_{tot} = \Delta E_{ref} + \frac{4m - n}{18} E_{tot}^* + \frac{n - m}{72} E_{tot}^{**} \quad (6)$$

where $E_{tot}^*$ and $E_{tot}^{**}$ are the DFT-computed absolute energies of benzene and circumcircumcoronene, respectively.

In the dataset for this general model, most of the planar benzenoid PAHs are drawn from the datasets in our previous study[28], supplemented by an appreciable number of additional molecules to ensure complete coverage of all even numbers of carbon atoms from 6 to 96. The resulting dataset of legitimate molecules comprises 2275 planar benzenoid PAHs with DFT-confirmed closed-shell ground states. Notably, this dataset excludes all molecules from the $C_{48}H_{24}$ dataset used in the preceding section, enabling us to assess the generalization ability of the general model for the $C_{48}H_{24}$ cases, as shown in the Results section.

Figure 8a demonstrates that the general autodesmotic model achieves excellent performance ($R^2 > 0.997$ and RMSE < 0.5 kcal/mol) in predicting the DFT-computed reference reaction energies for all considered PAHs from $C_6H_6$ to $C_{96}H_{24}$. The accuracy of the $C_{48}H_{24}$-specific model in predicting total energies of $C_{48}H_{24}$ PAHs is also verified in Supplementary Fig. S7a. All optimized model parameters are provided in Supplementary Note 9.

## Bond length prediction model

Given solely the topological information (connectivity between carbon atoms) of a reference PAH molecule, we aim to predict its equilibrium C–C bond lengths. The C–C bond length is found to be approximately proportional to the logarithm of its bond order[65], reflecting the exponential nature of interatomic forces. To enhance prediction accuracy, we further incorporate the local topological environment surrounding each bond by including information from its neighboring bonds and rings. Accordingly, we propose the following model for predicting the equilibrium C–C bond lengths in a given reference PAH molecule:

$$d_i = -\rho \ln(b_i) + \sum_{k \in \mathcal{T}} w_k t_{k,i} + \sum_{k \in \mathcal{R}} u_k \tau_{k,i} \quad (7)$$

Here, $d_i$ is the length of bond $i$; $b_i$ is its Coulson bond order from the HMO theory; and $\rho$ is a positive coefficient. The second term is a weighted sum over atom-based bond types indexed by the set $\mathcal{T}$ (illustrated in Supplementary Fig. S8), where each bond is assigned to exactly one type using mutually exclusive one-hot indicator variables $t_{k,i}$. Specifically, $t_{k,i} = 1$ if bond $i$ belongs to atom-based type $k$, otherwise 0. The third term similarly accounts for ring-based bond types defined by the set $\mathcal{R}$ (see Supplementary Fig. S9), with each type likewise encoded by mutually exclusive one-hot variables $\tau_{k,i}$. The weights $w_k$ and $u_k$ modulate the influence of atom- and ring-level environments, respectively, on bond length. This model thus integrates electronic structure information (bond order) with environmental topological features to enable accurate predictions of equilibrium bond lengths. A detailed explanation of the atom- and ring-based bond types is provided in Supplementary Note 10.

Unlike the energy prediction model employing the simple HMO method, the computation of Coulson bond orders in Eq. (7) is based on the distance-dependent[63] HMO approach. In this method, the resonance integral $\beta_i$ for the $i$th C–C bond is modeled as a function of the corresponding

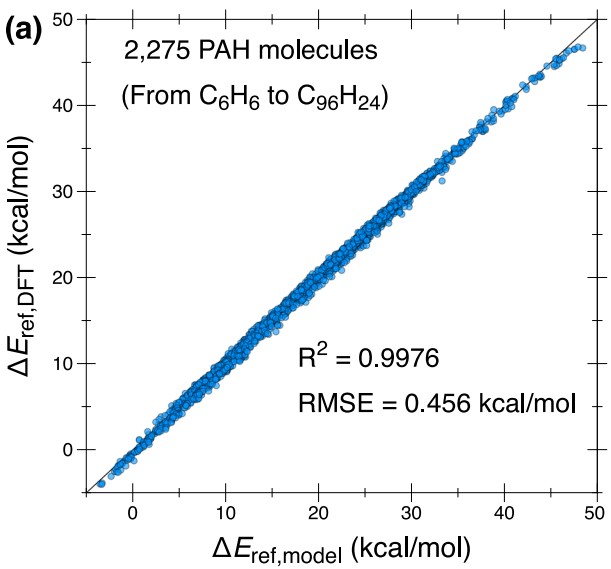

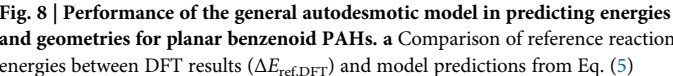

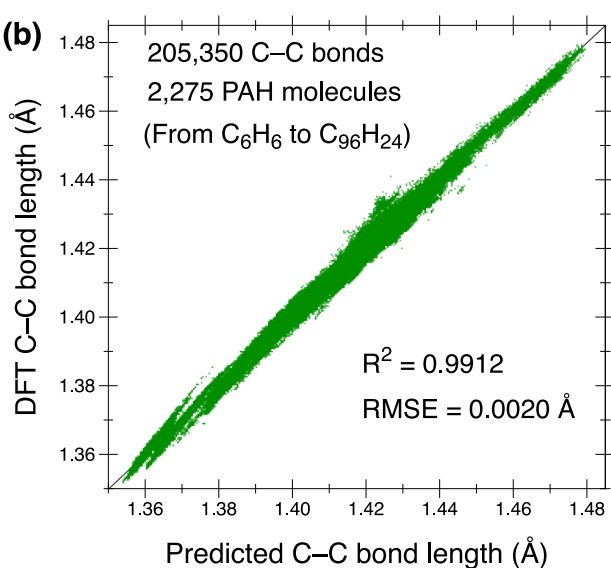

**Fig. 8 | Performance of the general autodesmotic model in predicting energies and geometries for planar benzenoid PAHs. a** Comparison of reference reaction energies between DFT results ($\Delta E_{\mathrm{ref,DFT}}$) and model predictions from Eq. (5) ($\Delta E_{\mathrm{ref,model}}$). **b** Comparison of C–C bond lengths between DFT calculations and model predictions (Eq. (7)). The dataset includes 2275 planar benzenoid PAHs from $C_6H_6$ to $C_{96}H_{24}$, comprising a total of 205,350 C–C bonds.

bond length $d_i$[63]. Specifically, we adopt a power-law form, $\beta_i = \beta(d_i/d^*)^\zeta$, where $d^* = 1.397$ Å is the equilibrium C–C bond length in benzene, and the exponent $\zeta$ is an empirical parameter optimized by fitting to DFT-computed bond lengths $\{d_i\}$. The Coulomb integral $\alpha$ is held constant for all carbon atoms, a reasonable assumption justified by the fact that the polyhex graphs of all reference PAHs are alternant, resulting in vanishing Hückel atomic charges according to the Coulson–Rushbrooke pairing theorem[66]. Coulson bond orders are then obtained from the modified adjacency matrix, constructed by replacing each nonzero element (originally 1) in the standard adjacency matrix with the bond-length-dependent factor $(d_i/d^*)^\zeta$.

The fitted parameter values in Eq. (7) are listed in Supplementary Note 9. Figure 8b shows good agreement between the model-predicted C–C bond lengths and those from DFT optimizations for 205,350 C–C bonds in 2275 reference PAHs of diverse sizes ($R^2 = 0.991$, RMSE = 0.002 Å). The model specific for $C_{48}H_{24}$ PAHs maintains similarly strong performance (see Supplementary Fig. S7b). To further validate the robustness and generalizability of the model, we tested its accuracy on ten large CNBs, each consisting of 120 rings, constructed by repeating the unit structure of the benchmark [12]CNBs shown in Fig. 3a. Because of their substantial sizes, these extended [120]CNBs closely approximate the infinite-size, strain-free limit. Importantly, these molecules were not included in the training dataset. As shown in Supplementary Fig. S10, the model retains satisfactory predictive accuracy, with $R^2 \approx 0.995$ and RMSE $\approx 0.002$ Å.

## SE evaluation from autodesmotic reactions

With the virtual chemical space established from the model fitting described above, the equilibrium C–C bond lengths and total energy of the autodesmotic reference can be interpolated by applying the same predictive models to the given strained molecule. Because the Coulson bond orders, $\{b_i\}$, and C–C bond lengths, $\{d_i\}$, are interdependent (Eq. (7)), an iterative self-consistent procedure is required to reach convergence. Specifically, the prediction of the reference's equilibrium C–C bond lengths is initialized using the bond lengths of the strained molecule, which determine the distance-dependent adjacency matrix for calculating HMO bond orders (Eq. (7)). The predicted bond lengths of the reference are then used to update the adjacency matrix, which in turn provides a refined prediction. This cycle is repeated until the equilibrium bond lengths converge, with the difference between successive iterations falling below $10^{-6}$ Å.

Subsequently, the reference's total energy is computed from its predicted equilibrium C–C bond lengths using Eq. (2) or Eqs (5) and (6). It is important to reiterate that, for energy predictions, we adopt the simple HMO scheme to maintain model parsimony and minimize the risk of overfitting, since incorporating distance-dependent HMO parameters yields negligible improvement in accuracy (Supplementary Figs. S11 and S12). In contrast, the distance-dependent HMO approach is employed for bond length predictions, where it captures subtle structural variations more reliably. Although the simple HMO model generally performs adequately, it produces significant errors in SE estimates for the Vögtle belt and its Möbius isomer[29] (see Supplementary Note 12). This difference arises from the fact that the distance-dependent HMO model more accurately describes bond lengths in conjugated systems, as evidenced by its correct prediction of bond length alternation in linear polyenes[63], whereas the simple HMO model fails to reproduce this feature[33].

Importantly, we also note that when the simple or distance-dependent HMO theory is applied to $\pi$-conjugated molecules with a Möbius topology, phase inversion of the $\pi$ atomic orbitals should be appropriately considered in both the construction of the Hamiltonian matrix and the calculation of Coulson bond orders[67].

## Quantum chemical calculations

All DFT calculations were conducted using the GAUSSIAN 16 software[68] at the B3LYP/6-311G*//B3LYP/6-31G* [69] level. Previous assessments[28] have shown that this level of theory provides a description of electron delocalization in PAHs comparable to that obtained with several range-separated functionals. The B3LYP functional has also been widely employed to investigate the SEs of various aromatic hydrocarbons, such as CNBs[19,29,30], curved PAHs[45], and fullerenes[31,56]. All geometries were fully optimized without constraints, and most were subsequently confirmed as true potential energy minima by vibrational frequency analysis. We adopted the widely used spin-unrestricted broken-symmetry DFT (UBS-DFT) approach[70,71] at the B3LYP/6-311G* level for all generated PAHs to determine whether each molecule exhibits a closed-shell, open-shell singlet, or triplet ground state. For each calculation, wave function stability was verified against RHF → UHF instabilities. In the full dataset, the singlet–triplet energy gap is at least 9 kcal/mol and the HOMO–LUMO gap exceeds 1.1 eV, indicating that the PAHs in the dataset are highly likely to possess closed-shell ground states. By using the ORCA program (version 6.1.0)[72], additional complete active space self-consistent field (CASSCF) and second-order N-

electron valence state perturbation theory (NEVPT2) calculations were performed for representative PAHs and further validated the UBS-DFT ground-state diagnosis (see Supplementary Note 13 for details).

SE calculations using StrainViz[29] were performed at the B3LYP/6-31G* level. For each segment along the macrocycle of a CNB, we considered the removal of a $C_{10}H_6$ piece (one benzenoid ring with four peripheral carbon atoms), with the addition of four hydrogen atoms to saturate the dangling bonds in the remaining fragment. This fragment retains complete benzenoid rings and is as large as possible, thereby minimizing the degree of structural approximation in the StrainViz method[29]. As test cases, our StrainViz calculations yielded SEs of 107 and 239 kcal/mol for the Vögtle belt and its Möbius form[29], respectively, closely reproducing the values of 105 and 238 kcal/mol reported in the original StrainViz paper[29].

## Data availability

The Supplementary Information provides additional details supporting the results and discussion in the main text. Cartesian coordinates and absolute energies of all molecules used for SE evaluation are included in Supplementary Data 1 (.zip). All data underlying the graphs in Figs. 3, 4, 6, 7 and 8 are provided in Supplementary Data 2–6 (.xlsx). Datasets for model training and validation have been deposited in a DOI-minting repository to ensure long-term accessibility and reproducibility. The GitHub repository has been archived on Zenodo, providing a permanent, citable release of the exact version used in this study[73]: https://doi.org/10.5281/zenodo.17732801. All other relevant data are available from the authors upon request[73].

## Code availability

The MATLAB implementation of the autodesmotic method developed in this study is publicly available on GitHub at https://github.com/yangwangmadrid/Autodesm, and has been archived on Zenodo to provide a permanent, citable release of the exact version used in this study[73]. The repository contains all essential components for reproducibility and reuse, including (i) source code for building and applying autodesmotic models, (ii) datasets used for model training and validation, (iii) a compiled standalone version of the AUTODESM software, and (iv) test examples demonstrating code usage. The input variables, model parameters, and datasets used to generate the results in this study are included. Computations were performed using MATLAB R2023. The code and datasets are fully accessible without restrictions.

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

## Acknowledgements
Y.W. acknowledges the financial support from National Natural Science Foundation of China (22073080, 22473097).

## Author contributions
Y.W. is the sole author of this article and performed all aspects of the work, including conceptualization, execution, analysis, and manuscript writing.

## Competing interests
The author declares no competing interests.

## Inclusion and ethics statement
As this is a sole-author computational study, there are no ethical or inclusion-related concerns. The author meets the criteria for authorship as defined by Nature Portfolio. No human or animal subjects were involved, and all data were generated and analyzed by the author. No external datasets or software were used without proper attribution. There are no ethical or inclusion concerns to declare.
