## [Transparent Peer Review file · Communications Chemistry]

Autodesmotic reactions for general strain energy evaluation in polycyclic aromatic nanocarbons

Corresponding Author: Professor Yang Wang

Version 0:

Reviewer comments:

Reviewer #1

(Remarks to the Author)

Wang presents a comprehensive paper on autodesmotic reactions, a novel strain quantification scheme for strained polycyclic aromatic hydrocarbons (PAHs). By mapping energetic and geometric quantities of a strained PAH onto a set of strain-free analogues, the strain energy of the PAH can be calculated at the cost of a single DFT calculation. This makes the method much more economical from a computational point of view than earlier approaches. Most importantly, however, autodesmotic reactions can be used in cases where previous schemes, such as isodesmic reactions or more expensive computational methods, are not applicable. This versatility makes autodesmotic reactions a useful tool to calculate strain energies of diverse systems such as carbon nanobelts, helicenes, nanotubes and fullerenes.

The paper is written in an exceptionally clear way and the author's claims are adequately backed by the presented data.

I recommend publication of the paper once the following issues have been resolved:

- 1) The definition of the parameter η (p. 22) remains unclear. What is its exact value? How was it determined? And why does it not depend on the precise value of the H...H close contact?
- 2) Supplementary note 5 mentions: "Finally, single-point B3LYP/6-311G* calculations verified that all these reference molecules possess closed-shell ground states." How can closed-shell ground states be identified with B3LYP/6-311G*? DFT often fails to account for low-lying triplet states in even small, unstrained PAHs, and is not a reliable tool to diagnose closed-shell or open-shell character. How was the diagnosis performed exactly? And why with DFT?
- 3) Besides calculation of the strain energy, localization of the most strained regions is very useful when discussing strained hydrocarbons, because it allows identifying reactive sites. Isodesmic and homodesmotic reactions sometimes advertise that they can do this, and StrainViz has proven its ability in this regard. Is this also possible with autodesmotic reactions?
- 4) How can the readers access the program? Is it available on github? In times of the FAIR principle, I strongly recommend making the code publicly available, which will not only enable others to reproduce the data, but it will also enable the widespread use of autodesmotic reactions.
- 5) Minor point: In the caption of Fig. S5, there are two "(b)". The second one should be a (d).

Reviewer #2

(Remarks to the Author)

This paper reports a new method for determining the strain energy (SE) of polycyclic aromatic hydrocarbons (PAHs). Quantifying strain energy is essential for evaluating the synthetic feasibility, stability, and reactivity of curved PAHs. Although various methods for calculating SE have been used to date, each has its own limitations. The biggest challenge is the dependency of SE on the reference compounds. SE inevitably varies significantly depending on the planar PAHs selected as the reference compounds, and there are no established criteria for selecting the reference compounds. The author developed a method that constructs a virtual chemical space from a vast number of planar PAHs and proposes reference values corresponding to the target curved PAH. If this method becomes a standard and is adopted by all PAH chemists, the reference compound-dependent problem will be resolved.

For the above reasons, this paper is of quality suitable for publication in Communications Chemistry. However, there are several concerns with this paper that need to be addressed.

1) The authors need to develop this method into a tool and make it available to synthetic chemists. StrainViz is currently widely used because it is an easy-to-use platform even for non-computational chemists. However, the methodology in this paper is complicated, making it extremely difficult to reproduce the methodology based solely on the paper and supporting information.

2) Because the chemical space is vague, it is unclear what the calculated SE is being compared to. It is feared that it will be necessary to compare the SE values with StrainViz to confirm that they are reasonable. Is it possible to suggest specific reference compounds when calculating the SE of a certain curved PAH?

3) This method is likely to take into account the strain of non-six-membered rings, such as four- and five-membered rings. On the other hand, π -conjugated units such as biphenylene and dibenzo[a,e]pentalene are synthetically strong, so synthetic chemists are not interested in the energy required for the cleavage of their four- or five-membered rings. In this regard, StrainViz can calculate the SE of the bond of interest, and π -conjugated units such as biphenylene and dibenzopentalene can be excluded from the SE calculation as a block. The authors should apply this method to bent PAHs, including biphenylene and dibenzopentalene, and report the results.

4) Currently, the method's applicability is limited to molecules consisting only of sp² carbons and hydrogens. Its usefulness is significantly reduced compared to StrainViz, which can also be applied to carbons with other hybridizations and heteroatoms. The authors should present a solution to this issue in their future work.

5) Page 5: "This method is inapplicable to nonperiodic CNBs" is incorrect. Since all CNBs have their integer multiples, the SE of nonperiodic CNBs can be calculated.

6) Fig 4: While the StrainViz and Autodesmotic methods optimize the (2,2) CNB itself to calculate the SE, the Asymptotic method extrapolates the predicted value from the larger CNB, which is unfair. For example, the following wording should be added: "This is an extrapolated reference value for the asymptotic method only, without the optimization of (2,2)CNB. It is presumed that it predicts the SE of (2,2)CNB, which maintains the aromaticity of the benzene ring, rather than the stable quinoidal structure."

Reviewer #3

(Remarks to the Author)

This manuscript provides an interesting next step in the evolution of methodology towards estimating strain energy. The method constructs a reference that meets five reasonable criteria for defining an unstrained molecule through a machine learning approach. The method provides strain energies consistent with other methods, and affords strain energies for molecules that are difficult if not impossible to evaluate using other methods.

I recommend that the manuscript be published as it represents a good improvement, but I do suggest the author consider some caveats.

1) Every method proposed in the past or in the future will by definition be arbitrary. There is no "strain energy operator" within quantum mechanics. Further the definition of strain energy assumes some unstrained reference. One cannot get away from this.

2) This method has its arbitrary factors too. The set of reference molecules, the defining energy equation (#1) and the equations used to define each of the energy components.

3) The computational savings is in the eye of the beholder. Yes if one is computing the strain using some homodesmotic like approach, in addition to the molecule in question, all of the reference molecules must be computed, but if one is looking at a large number of molecules, then many of the reference compounds are being used over and over again and do not need to be recomputed. And what about the computational cost in creating the autodesmotic references? And if one wanted to look at say nitrogen substituted CNB, then a whole lot of reference molecules will need to be computed to create the new model – and that might also identify some systemic missing pieces to the defining equations....

So, what I would like to see is the author recognizing these inherent limitations in this, and in any model, and recognize that no one method will ever be truly unique and without need of some correction. (In fact, the author already admits this last point in the discussion involving kekulene, but does not amplify its implication.)

Version 1:

Reviewer comments:

Reviewer #1

(Remarks to the Author)

The author has addressed all my comments.

Reviewer #2

(Remarks to the Author)

The author clearly acknowledge the limitations of the proposed method (local strains, non-hexagonal systems) and suggest future research directions to address these issues, which is satisfying.

This reviewer believes this paper is of sufficient quality for publication in Communications Chemistry and strongly recommend its acceptance.

Reviewer #3

(Remarks to the Author)

This reviewer (#3 in the previous pass) has looked over the changes to the manuscript that address the points I made. I am satisfied with these additions and changes.

The manuscript is now acceptable for publication as is.

Reviewers' comments:

Reviewer #1 (Remarks to the Author):

Wang presents a comprehensive paper on autodesmotic reactions, a novel strain quantification scheme for strained polycyclic aromatic hydrocarbons (PAHs). By mapping energetic and geometric quantities of a strained PAH onto a set of strain-free analogues, the strain energy of the PAH can be calculated at the cost of a single DFT calculation. This makes the method much more economical from a computational point of view than earlier approaches. Most importantly, however, autodesmotic reactions can be used in cases where previous schemes, such as isodesmic reactions or more expensive computational methods, are not applicable. This versatility makes autodesmotic reactions a useful tool to calculate strain energies of diverse systems such as carbon nanobelts, helicenes, nanotubes and fullerenes.

The paper is written in an exceptionally clear way and the author's claims are adequately backed by the presented data.

I recommend publication of the paper once the following issues have been resolved:

Author reply: We thank the reviewer for the positive and encouraging evaluation. We have carefully considered all comments and addressed each point in detail below.

1) The definition of the parameter η (p. 22) remains unclear. What is its exact value? How was it determined? And why does it not depend on the precise value of the H...H close contact?

Author reply: We thank the reviewer for pointing out the need for clarification. In our model, the parameter η represents *an average repulsion energy per close H...H contact*, introduced to approximate the cumulative effect of short H...H contacts (if present) in the PAHs across the dataset. Because η is an empirical mean parameter derived from a diverse set of molecules, its purpose is not to describe the detailed distance dependence of a specific H...H pair interaction, but rather to provide a coarse-grained measure of overall H...H steric repulsion in PAHs. Therefore, η is treated as a constant factor instead of a distance-dependent function. The fitted value of η from the 1,516 C₄₈H₂₄ PAHs is about 0.89 kcal/mol (Page 16 in the Supplementary Information), while a similar value of 1.38 kcal/mol is obtained from fitting the broader dataset with 2,275 PAHs from C₆H₆ to C₉₆H₂₄ (Page 19, Supplementary Information). These η values are consistent with previous estimate of H...H repulsion energy (roughly 1 kcal/mol) inferred for phenanthrene (see the discussion on Page 16 and refs. 35, 36 in the Supplementary Information).

To evaluate this approximation, we also tested a variant model that explicitly computes pairwise H...H repulsion energies using either a Lennard–Jones or a Buckingham potential. For the dataset comprising 2,275 PAH molecules, these distance-dependent potentials showed no improvement or only a very slight improvement in prediction accuracy compared with the original coarse-grained model using a single η parameter. The squared correlation coefficients (R^2) were 0.9976, 0.9974, and 0.9985, and the RMSE values were 0.456, 0.472, and 0.361 kcal/mol for the coarse-grained, Lennard–Jones, and Buckingham models, respectively.

More importantly, the use of distance-dependent potentials introduces additional fitting parameters, which often leads to numerical instabilities and multiple (sometimes physically unmeaningful) minima during model optimization. Furthermore, over-parameterization can reduce the model's generalization ability. For these reasons, we prefer the coarse-grained model (i.e., using a single parameter η to account for average H \cdots H repulsion), which keeps the model simple yet effective, numerically stable, and more transferable.

Accordingly, in the revised manuscript (Page 24), we have enhanced the definition of η as “*where η denotes an empirical parameter representing the average repulsion energy per close H \cdots H contact*”.

We have also added the following sentences to further explain this choice (Page 25):

“This coarse-grained treatment of overall H \cdots H steric repulsion keeps the model simple yet effective, numerically stable, and broadly transferable. Tests show that incorporating explicit pairwise H \cdots H repulsion using Lennard–Jones or Buckingham potentials yields no improvement, or only marginal improvement, over the coarse-grained model. Furthermore, such distance-dependent potentials introduce additional fitting parameters, which reduce numerical stability in model optimization and compromise model generalizability.”

2) Supplementary note 5 mentions: “Finally, single-point B3LYP/6-311G* calculations verified that all these reference molecules possess closed-shell ground states.” How can closed-shell ground states be identified with B3LYP/6-311G*? DFT often fails to account for low-lying triplet states in even small, unstrained PAHs, and is not a reliable tool to diagnose closed-shell or open-shell character. How was the diagnosis performed exactly? And why with DFT?

Author reply: In this work, we have adopted the spin-unrestricted broken-symmetry DFT (UBS-DFT) approach for open-shell calculations. Given the large number (> 2,000) and broad size range (up to C₉₆H₂₄) of molecules in our dataset, it is impractical to perform the computationally very demanding multiconfigurational ab initio calculations. The UBS-DFT method has been widely applied to diradical and polyradical PAHs, with scattered examples including *J. Phys. Chem. A* **2008**, *112*, 332; *PCCP* **2011**, *13*, 20575; *Nat. Chem.* **2011**, *3*, 197; *Commun. Chem.* **2022**, *5*, 127; *Angew. Chem. Int. Ed.* **2023**, *62*, e202309238; *JACS*, **2024**, *146*, 38, 26454; *Phys. Rev. B* **2025**, *112*, 094404. It has been shown to yield reasonable results, supported by high-level multiconfigurational methods such as CASSCF and MR-MP2 (*JACS* **2004**, *126*, 7416; *Org. Lett.* **2007** *9*, 5449; *J. Mol. Model.* **2011**, *17*, 805). In particular, hybrid functionals (such as B3LYP employed in this work) are shown to outperform pure DFT functionals, as they mitigate self-interaction error and better capture static electron correlation (*Int. J. Mol. Sci.* **2002**, *3*, 360).

Specifically, we carried out large-scale UBS-DFT calculations at the B3LYP/6-311G* level for all generated PAHs to identify whether each molecule possesses a closed-shell, open-shell singlet, or triplet ground state. In each calculation, the wave function stability was verified against RHF→UHF instabilities using the STABLE=Opt keyword in the Gaussian 16 program.

To further validate our UBS-DFT results, we have, during the preparation of this revision, selected several representative PAHs of moderate size (for the sake of computational feasibility) and

performed additional CASSCF and NEVPT2 single-point calculations. The chemical structures of these test PAHs are shown in Supplementary Fig. S16 (Page 29). Specifically, state-averaged (SA) CASSCF(10,10) calculations were carried out over two singlet and two triplet roots. Our choice of active space sizes appears reasonable since in a recent study CAS(8,8) reference spaces were used to compute the lowest singlet/triplet states of comparably sized PAHs exhibiting polyradical character (*PCCP* **2023**, 25, 27380).

Table R1. Comparison of SA-CASSCF(10,10), NEVPT2 and UBS-DFT results for representative PAHs in the dataset.

Molecule	Ground state	$\Delta E(\text{T-S})^a$ (kcal/mol)			c_1 and c_2 (CASSCF) ^b
		CASSCF	NEVPT2	UBS-DFT	
coronene	Closed-shell	81.8	74.2	67.1	0.88094, 0.01411
$\text{C}_{28}\text{H}_{14}^c$	Closed-shell	32.4	23.4	18.8	0.86421, 0.03362
hexacene	Open-shell singlet	18.6	18.0	12.8	0.62117, 0.12692
$\text{C}_{28}\text{H}_{16}^d$	Open-shell singlet	24.9	20.2	12.1	0.76541, 0.07626
$\text{C}_{32}\text{H}_{18}^d$	Open-shell singlet	11.0	12.3	7.1	0.68402, 0.16192
triangulene	Triplet	-20.9	-12.5	-7.1	-
$\text{C}_{30}\text{H}_{16}^d$	Triplet	-20.3	-12.0	-5.6	-

^aEnergy difference between the lowest triplet and singlet states, defined as $E(\text{triplet}) - E(\text{singlet})$.

^bConfiguration interaction (CI) coefficients for the two leading configuration state functions (CSFs) in the ground-state CASSCF wave function.

^cPhenanthro[1,10,9,8-opqra]perylene; see Fig. S16 for its chemical structure.

^dSee Fig. S16 for their chemical structures.

As shown in Table R1, the UBS-DFT approach provides a qualitatively reliable ground-state diagnosis for the systems studied. The singlet–triplet energy gaps, $\Delta E(\text{T-S})$, predicted by UBS-DFT are of the same order of magnitude as those obtained from NEVPT2 calculations. For coronene and phenanthro[1,10,9,8-opqra]perylene ($\text{C}_{28}\text{H}_{14}$), the CI coefficient of the leading CSF (> 0.86) is substantially larger than those of minor CSFs (< 0.04), indicating that these molecules possess closed-shell ground states. In contrast, for hexacene, $\text{C}_{28}\text{H}_{16}$, and $\text{C}_{32}\text{H}_{18}$, the leading CI coefficient decreases significantly (0.62–0.76), suggesting pronounced diradical/polyradical character. These results are consistent with the UBS-DFT ground-state diagnosis.

After all, one of the primary objectives of this study is to construct a virtual chemical space comprising a large number of PAHs with closed-shell ground states. Accordingly, our main concern is to ideally avoid false positives in the dataset, that is, to ensure that no open-shell species are mistakenly included; in contrast, the exclusion of a true closed-shell molecule misdiagnosed as an open-shell singlet or triplet (i.e., a false negative) has no practical consequence for the reliability of the dataset. To this end, we examined the singlet–triplet energy gaps and HOMO–LUMO gaps obtained at the UBS-DFT level for all 2,275 PAHs in our dataset. In every case, the singlet–triplet energy gap is at least 9 kcal/mol and the HOMO–LUMO gap exceeds 1.1 eV. These appreciable values indicate that the PAHs included in our dataset are highly likely to possess closed-shell ground states.

In the “Methods” section of the revised manuscript (Pages 30–31), we have clarified the procedure used to identify ground states of all generated PAHs and justified the use of the UBS-DFT approach by adding the following text:

“We adopted the widely used spin-unrestricted broken-symmetry DFT (UBS-DFT) approach^{70,71} at the B3LYP/6-311G level for all generated PAHs to determine whether each molecule exhibits a closed-shell, open-shell singlet, or triplet ground state. For each calculation, wave function stability was verified against RHF→UHF instabilities. In the full dataset, the singlet–triplet energy gap is at least 9 kcal/mol and the HOMO–LUMO gap exceeds 1.1 eV, indicating that the PAHs in the dataset are highly likely to possess closed-shell ground states. By using the ORCA program (version 6.1.0),⁷² additional complete active space self-consistent field (CASSCF) and second-order N-electron valence state perturbation theory (NEVPT2) calculations were performed for representative PAHs and further validated the UBS-DFT ground-state diagnosis (see Supplementary Note 13 for details).”*

In the revised Supplementary Information (Pages 29–30), we have added Supplementary Note 13, entitled “*Validation of Ground-State Diagnosis by UBS-DFT*”, which provides the detailed multireference validation summarized above. Table S5 therein reproduces Table R1 presented above.

3) Besides calculation of the strain energy, localization of the most strained regions is very useful when discussing strained hydrocarbons, because it allows identifying reactive sites. Isodesmic and homodesmotic reactions sometimes advertise that they can do this, and StrainViz has proven its ability in this regard. Is this also possible with autodesmotic reactions?

Author reply: We agree that identifying and quantifying local strain energies in different regions of a molecule is of great chemical value. Unfortunately, it remains methodologically very challenging to decompose the total strain energy into contributions from specific bonds, bond angles, or torsional angles, due to the intrinsic structure of the autodesmotic framework. Please allow us to explain this point in detail below.

As shown in Eqs. (1) and (2), the total energy of an ideal, strain-free reference PAH can be decomposed into contributions from the π -conjugated system, the C–C σ bonds, the repulsion between close H···H contacts, and a constant term representing all remaining energies. While the energies of C–C σ bonds and H···H repulsions can be unambiguously assigned to individual bonds and contacts, the π -conjugation energy cannot be rigorously partitioned into bond-wise contributions because the π electrons are inherently delocalized. In Hückel molecular orbital (HMO) theory, the total π energy is calculated for the π system as a whole. One may come up with approximate partitioning schemes: for example, assigning bond energies proportional to their π bond orders (or to the logarithm of their π bond orders). However, these schemes remain heuristic, and their reliability requires further validation.

Moreover, such π -energy decomposition, as well as the π -energy expression in Eqs. (1) and (2), is applicable only to *planar* PAHs. In nonplanar strained molecules, geometric distortions reduce the overlap between π atomic orbitals, invalidating the prerequisite assumptions of HMO theory. In the autodesmotic model, we therefore do not apply HMO theory or Eqs. (1) and (2) to *strained* PAHs. Instead, we use only the DFT-computed total energy of the strained molecule and the

model-predicted total energy of its corresponding hypothetical reference (via Eqs. (1) and (2)) to evaluate the total strain energy.

Therefore, this intrinsic methodological limitation explains why assigning local strain energy to specific regions is not currently possible within the present autodesmotic framework. Overcoming this limitation is an important and interesting challenge that could significantly improve the current method, but it is a nontrivial task that would require innovative approaches and lies beyond the scope of the present work.

In response to the reviewer's comment, we have added the following discussion in the revised manuscript (penultimate paragraph in the "Discussion" section, Page 23):

"Finally, while the current framework quantifies total SE globally for the entire molecule, it does not yet allow decomposition into specific bonds, angles, or regions, for which StrainViz has proven effective. This limitation results from the delocalized nature of π electrons, which prevents rigorous bond-wise partitioning of π -energy. Developing strategies for spatially resolved SE decomposition is therefore an important methodological challenge and an intriguing direction for future work."

4) How can the readers access the program? Is it available on github? In times of the FAIR principle, I strongly recommend making the code publicly available, which will not only enable others to reproduce the data, but it will also enable the widespread use of autodesmotic reactions.

Author reply: We thank the reviewer for highlighting the importance of making our code publicly available to support reproducibility of this work and facilitate broader use of the autodesmotic method in the community. We have now deposited a MATLAB implementation of the autodesmotic method on GitHub, together with all required data and usage instructions:

- Source code:
<https://github.com/yangwangmadrid/Autodesm>
- Datasets for building autodesmotic models:
<https://github.com/yangwangmadrid/Autodesm/tree/main/Datasets>
- Compiled standalone version of the software:
https://github.com/yangwangmadrid/Autodesm/tree/main/Models_App
- Test examples demonstrating code usage:
https://github.com/yangwangmadrid/Autodesm/tree/main/Test_Cases
- README with detailed instructions:
<https://github.com/yangwangmadrid/Autodesm/blob/main/README.md>

In the revised manuscript (Page 31), we have added a new "Code availability" section that specifies how to access the code, datasets, and test examples of our method.

5) Minor point: In the caption of Fig. S5, there are two "(b)". The second one should be a (d).

Author reply: The typo has been corrected in the revised Supplementary Information (Page 11).

Reviewer #2 (Remarks to the Author):

This paper reports a new method for determining the strain energy (SE) of polycyclic aromatic hydrocarbons (PAHs). Quantifying strain energy is essential for evaluating the synthetic feasibility, stability, and reactivity of curved PAHs. Although various methods for calculating SE have been used to date, each has its own limitations. The biggest challenge is the dependency of SE on the reference compounds. SE inevitably varies significantly depending on the planar PAHs selected as the reference compounds, and there are no established criteria for selecting the reference compounds. The author developed a method that constructs a virtual chemical space from a vast number of planar PAHs and proposes reference values corresponding to the target curved PAH. If this method becomes a standard and is adopted by all PAH chemists, the reference compound-dependent problem will be resolved.

For the above reasons, this paper is of quality suitable for publication in *Communications Chemistry*. However, there are several concerns with this paper that need to be addressed.

Author reply: We sincerely thank the reviewer for the positive and thoughtful comments, as well as the constructive suggestions. Each point has been carefully considered and addressed below.

1) The authors need to develop this method into a tool and make it available to synthetic chemists. StrainViz is currently widely used because it is an easy-to-use platform even for non-computational chemists. However, the methodology in this paper is complicated, making it extremely difficult to reproduce the methodology based solely on the paper and supporting information.

Author reply: We thank the reviewer for pointing out this important issue. As a similar comment was made by Reviewer #1, we provide the same response here for clarity:

We thank the reviewer for highlighting the importance of making our code publicly available to support reproducibility of this work and facilitate broader use of the autodesmotic method in the community. We have now deposited a MATLAB implementation of the autodesmotic method on GitHub, together with all required data and usage instructions:

- Source code:

<https://github.com/yangwangmadrid/Autodesm>

- Datasets for building autodesmotic models:

<https://github.com/yangwangmadrid/Autodesm/tree/main/Datasets>

- Compiled standalone version of the software:

https://github.com/yangwangmadrid/Autodesm/tree/main/Models_App

- Test examples demonstrating code usage:

https://github.com/yangwangmadrid/Autodesm/tree/main/Test_Cases

- README with detailed instructions:

<https://github.com/yangwangmadrid/Autodesm/blob/main/README.md>

In the revised manuscript (Page 31), we have added a new “Code availability” section that specifies how to access the code, datasets, and test examples of our method.

2) Because the chemical space is vague, it is unclear what the calculated SE is being compared to. It is feared that it will be necessary to compare the SE values with StrainViz to confirm that they are reasonable. Is it possible to suggest specific reference compounds when calculating the SE of a certain curved PAH?

Author reply: We appreciate the reviewer's concern regarding the reference compounds for SE evaluation. The short answer is that in the autodesmotic framework, there is no need to select specific reference compounds for a given curved PAH.

In conventional isodesmotic or homodesmotic reaction schemes, the calculated SE indeed depends strongly on the arbitrarily chosen reference compounds. The autodesmotic model was specifically designed to minimize this arbitrariness. In this framework, the strain-free reference structure is generated *automatically* through self-reaction interpolation within an operationally well-defined chemical space, without requiring any externally defined reference molecules. The SE is therefore obtained by comparing the energy of the target curved PAH with that of its uniquely defined, hypothetical strain-free reference.

The underlying "chemical space" is not arbitrary but systematically constructed or trained from a broad set of planar benzenoid PAHs. Within this space, the autodesmotic reference is a single-molecule, strain-free isomer that retains exactly the same atomic connectivity as the target curved PAH. Because the reference preserves the bonding topology, HMO theory implies that any difference in π -energy between the target and its reference originates solely from geometric or steric distortions, rather than from variations in π -bonding patterns dictated by atomic connectivity. In essence, the autodesmotic reference represents the *hypothetical unstrained counterpart* of the curved PAH, carrying the ideal π -energy that the curved PAH would possess if it could be perfectly planarized.

While such a strain-free reference may not exist as a real molecule in three-dimensional space, it is operationally defined within a higher-dimensional model space derived from known planar benzenoid PAHs via model fitting or machine learning. In the trivial limit of planar unstrained PAHs, the autodesmotic reference should coincide with the target molecule itself (within the model's fitting accuracy), indicating the internal consistency of the method.

This topology-preserving and self-consistent definition of the reference is the key conceptual advantage of the autodesmotic framework, allowing that SE values are directly comparable across diverse curved PAHs without relying on arbitrarily chosen external references, as required in conventional reference-based schemes.

Based on the above clarification, in the revised manuscript, we have added the following sentence on Page 7:

"As a result, any difference in π -energy between the target molecule and its reference originates effectively from geometric or steric distortions, rather than from variations in π -bonding patterns dictated by atomic connectivity."

At the end of the first paragraph in the "Discussion" section (Page 20), we have added the following text:

"In essence, the autodesmotic reference represents the hypothetical unstrained counterpart of the curved molecule, carrying the ideal π -energy that the curved molecule would possess if it could be perfectly planarized. Although such a strain-free reference may not exist as a real molecule in

three-dimensional space, it is operationally and reproducibly defined within the higher-dimensional chemical space, allowing that SE values are directly comparable across diverse curved molecules without reliance on arbitrarily chosen external references.”

3) This method is likely to take into account the strain of non-six-membered rings, such as four- and five-membered rings. On the other hand, π -conjugated units such as biphenylene and dibenzo[a,e]pentalene are synthetically strong, so synthetic chemists are not interested in the energy required for the cleavage of their four- or five-membered rings. In this regard, StrainViz can calculate the SE of the bond of interest, and π -conjugated units such as biphenylene and dibenzopentalene can be excluded from the SE calculation as a block. The authors should apply this method to bent PAHs, including biphenylene and dibenzopentalene, and report the results.

Author reply: We appreciate the reviewer’s thoughtful comment and insightful suggestions. We fully agree that, for synthetic chemists, local SE or relative SE referenced to intrinsically constrained building blocks (such as biphenylene and dibenzo[a,e]pentalene containing non-six-membered rings) is often more practically informative than the *absolute total* SE of the entire molecule. For instance, in the synthesis of CNBs containing pentagonal units (e.g., *Angew. Chem. Int. Ed.* **2025**, *64*, e202510544), the relevant energy buildup for macrocycle closure is measured relative to an already strained, pentagon-containing framework rather than to an idealized strain-free reference.

At the present stage, however, the autodesmotic approach is designed to evaluate the *total* SE of π -conjugated molecules. This is because the virtual reference chemical space underlying our model is constructed entirely from planar *benzenoid* PAHs composed exclusively of six-membered rings. Accordingly, autodesmotic references that treat four- or five-membered rings as intrinsically strained building blocks are not yet available. For the practical task suggested by the reviewer, namely, evaluating the SE of a bond of interest or relative SE by excluding π -conjugated units such as biphenylene and dibenzopentalene from the calculation, our current implementation cannot yet perform this operation. In this regard, StrainViz is indeed well suited, as it removes a small structural portion and relaxes the remaining fragment to its unstrained geometry to serve as a reference, thereby directly isolating the SE associated with macrocycle closure or a specific bond.

Nevertheless, the autodesmotic framework is conceptually general and can, in principle, be extended to evaluate relative SEs with respect to intrinsically strained references. This would require constructing an expanded virtual chemical space from non-benzenoid PAHs that include non-six-membered rings, thereby enabling the definition of autodesmotic references for molecules with mixed ring sizes. We envision this as a promising direction that would allow quantification of SE arising from framework bending or macrocycle closure for molecules containing specific non-six-membered-ring motifs. Such an extension would substantially broaden the applicability and chemical insight of the autodesmotic concept. We again appreciate the reviewer’s valuable suggestion and plan to explore this line in future work.

Based on the above discussion and clarification, we have added the following text in the revised manuscript (penultimate paragraph in the “Discussion” section, Page 22):

“The current autodesmotic method has several limitations. ... Second, the method evaluates SE only relative to six-membered benzenoid units, treating all deviations from hexagonal topology as

sources of strain. Expanding the model to include non-benzenoid PAHs containing four- and five-membered rings would enable autodesmotic references for molecules built from such units, allowing quantification of SE associated with framework bending and macrocycle closure in non-benzenoid systems. This extension provides more directly useful information for synthetic chemists.”

4) Currently, the method's applicability is limited to molecules consisting only of sp² carbons and hydrogens. Its usefulness is significantly reduced compared to StrainViz, which can also be applied to carbons with other hybridizations and heteroatoms. The authors should present a solution to this issue in their future work.

Author reply: We thank the reviewer for this insightful comment. As the reviewer correctly notes, the current formulation of the autodesmotic method is limited to π -conjugated molecules composed exclusively of sp²-hybridized carbon atoms. Consequently, StrainViz and the asymptotic methods have an advantage in handling nanobelts doped with heteroatoms or containing carbons of other hybridizations.

This limitation can, in principle, be overcome, although it would require careful and substantial further development. Extending the autodesmotic method to chemically diverse families is a natural and promising direction for future work. One strategy is to reformulate or augment the current models to predict energies and equilibrium geometries for heteroatom-doped or functionalized reference structures. A more general approach is to construct the virtual chemical space directly using data-driven machine-learning models, bypassing the need for explicit analytical forms in theoretical models. Both strategies would require the selection and computation of a sufficiently large set of representative model molecules to properly capture the chemical space of heteroatom-doped or functionalized references. While such extensions are beyond the scope of the present study, we believe they will substantially broaden the applicability and utility of the autodesmotic concept for a wider range of chemically complex nanocarbons. We plan to pursue this research direction in future work.

In the “Discussion” section of the revised manuscript (Page 22), we have added the following text to outline these possible extensions:

“The current autodesmotic method has several limitations. First, its implementation is restricted to fully conjugated hydrocarbons and carbon allotropes containing only sp² carbons. Extending the framework to heteroatom-doped or functionalized systems will require redefining the reference chemical space to include representative heteroatomic and mixed-hybridization motifs, or alternatively, training fully data-driven models that learn directly from diverse molecular datasets.”

5) Page 5: "This method is inapplicable to nonperiodic CNBs" is incorrect. Since all CNBs have their integer multiples, the SE of nonperiodic CNBs can be calculated.

Author reply: We agree with the reviewer on this point. We have removed the incorrect statement. The revised sentence now reads: *“This method may fail when other size-dependent effects (e.g., steric repulsion in helicene-containing CNBs³⁰) are significant, requiring non-trivial corrections.”* (Page 5 of the revised manuscript).

6) Fig 4: While the StrainViz and Autodesmotic methods optimize the (2,2) CNB itself to calculate the SE, the Asymptotic method extrapolates the predicted value from the larger CNB, which is unfair. For example, the following wording should be added: "This is an extrapolated reference value for the asymptotic method only, without the optimization of (2,2)CNB. It is presumed that it predicts the SE of (2,2)CNB, which maintains the aromaticity of the benzene ring, rather than the stable quinoidal structure."

Author reply: We thank the reviewer for pointing out this issue, which helps make our discussion more objective and rigorous. We agree that the large deviation of the asymptotic result for (2,2)CNB from StrainViz and autodesmotic methods is caused by its use of energy extrapolation rather than the direct optimization of (2,2)CNB. To clarify this point, in the main text (Page 13 of the revised manuscript), we have changed the original wording:

"This large deviation reflects the failure of the quadratic energy–size relationship for very small CNBs (see Supplementary Note 3)."

to the following:

"This large deviation is due to the fact that the SE predicted by the asymptotic method is an extrapolated reference value, obtained without optimizing [4]CNB (see Supplementary Note 3 for the detailed regression analysis). It is assumed to represent the SE of (2,2)CNB while preserving the aromaticity of the benzene rings, rather than adopting the stable quinoidal structure."

Reviewer #3 (Remarks to the Author):

This manuscript provides an interesting next step in the evolution of methodology towards estimating strain energy. The method constructs a reference that meets five reasonable criteria for defining an unstrained molecule through a machine learning approach. The method provides strain energies consistent with other methods, and affords strain energies for molecules that are difficult if not impossible to evaluate using other methods. I recommend that the manuscript be published as it represents a good improvement, but I do suggest the author consider some caveats.

Author reply: We sincerely thank the reviewer for the thoughtful and constructive evaluation of our work. We appreciate the reviewer's recognition of its significance and the valuable comments provided. The caveats raised have been explicitly acknowledged and discussed in the revised manuscript. Detailed responses to the individual comments are provided below.

1) Every method proposed in the past or in the future will by definition be arbitrary. There is no "strain energy operator" within quantum mechanics. Further the definition of strain energy assumes some unstrained reference. One cannot get away from this.

Author reply: We agree with the reviewer on this point. Indeed, strain energy (SE) is not an observable in quantum mechanics and any practical SE definition requires the choice of an unstrained reference. The contribution of our work is not to claim an ontologically unique SE operator, but to provide a transparent, physically motivated, and computationally efficient

operational definition that reduces a major source of ambiguity for π -conjugated nanocarbons: imbalance of π -energy between target and reference. By enforcing isotopological mapping (preserving atomic connectivity) of the target molecule, autodesmotic reactions mitigate this important class of arbitrariness (π -energy imbalance).

To address this and the following comment regarding arbitrariness, we have added a clarifying paragraph as a new second paragraph in the “Discussion” section (Pages 20–21), which explicitly explains the operational nature of SE, the sources of arbitrariness in the autodesmotic approach, and how isotopological mapping mitigates π -energy imbalance between target and reference systems. The concrete text of the added paragraph is provided at the end of our reply to the next comment.

2) This method has its arbitrary factors too. The set of reference molecules, the defining energy equation (#1) and the equations used to define each of the energy components.

Author reply: We fully agree with the reviewer that our method, like any practical definition of strain energy, inevitably involves certain arbitrary elements. We admit that the equations used to construct the virtual chemical space that defines the references are necessarily approximate. Nonetheless, the key advantage of the autodesmotic framework lies in that it provides a systematic and improvable way to mitigate an important source of arbitrariness in conventional approaches, namely, the π -energy imbalance between the target and reference systems. Although the current energy-model equations, while effective, describe the virtual chemical space with limited accuracy because of the approximations in their formulations and parameterization, the model can be further refined by employing more advanced regression or machine-learning techniques, offering a systematic route to improve accuracy and reduce arbitrariness. The present work should thus be viewed as a proof-of-concept demonstration of a generalizable framework for defining and evaluating strain energy in π -conjugated nanocarbons.

To clarify the issues raised in this and the preceding comment, we have added the following paragraph as a new second paragraph in the “Discussion” section (Pages 20–21):

“It is important to emphasize that SE is not a quantum-mechanical observable but an operationally defined quantity dependent on the chosen unstrained reference. Consequently, every method for estimating SE entails some degree of arbitrariness. In the autodesmotic approach, arbitrariness arises from the selection of training molecules used to construct the chemical space and from the analytical forms of the model equations. The aim of the present work is not to define a unique ‘strain energy operator’, but to minimize ambiguity in a physically transparent and computationally efficient manner. By enforcing isotopological mapping, which preserves the atomic connectivity of the target molecule in its reference, autodesmotic reactions mitigate a key source of arbitrariness in π -conjugated systems, namely, the imbalance of π -energy between target and reference. Moreover, the approximations in the present implementation can be systematically refined through re-parameterization or by adopting advanced machine-learning schemes, thereby reducing arbitrariness and improving predictive accuracy.”

3) The computational savings is in the eye of the beholder. Yes if one is computing the strain using some homodesmotic like approach, in addition to the molecule in question, all of the

reference molecules must be computed, but if one is looking at a large number of molecules, then many of the reference compounds are being used over and over again and do not need to be recomputed. And what about the computational cost in creating the autodesmotic references? And if one wanted to look at say nitrogen substituted CNB, then a whole lot of reference molecules will need to be computed to create the new model – and that might also identify some systemic missing pieces to the defining equations.... So, what I would like to see is the author recognizing these inherent limitations in this, and in any model, and recognize that no one method will ever be truly unique and without need of some correction. (In fact, the author already admits this last point in the discussion involving kekulene, but does not amplify its implication.)

Author reply: We thank the reviewer for these insightful and constructive comments. We agree that computational savings depend on context and on how reference data are organized and reused. Indeed, in some homodesmic like approaches, if one studies a large number of related molecules, many reference compounds can be reused to reduce the total cost. However, to the best of our knowledge, no general or rigorous framework currently exists that allows this reuse to be performed systematically across diverse π -conjugated nanocarbons while maintaining consistency in reference selection. In this regard, the autodesmotic framework proposed here provides such a systematic and generalizable route. By constructing all references within a unified virtual chemical space, the approach formalizes procedures that are otherwise handled in an ad hoc manner in homodesmic like approaches.

We acknowledge that the computational effort required to establish the autodesmotic reference space is substantial. However, this upfront investment follows a “once-and-for-all” logic analogous to the model-building effort following the machine-learning paradigm: once the virtual chemical space is trained and parameterized, each new molecule within the same chemical family requires only a single quantum-chemical calculation to evaluate its strain energy.

In the manuscript (Table 1), we compare the computational efficiency of the autodesmotic method specifically with the asymptotic and StrainViz methods. For these two existing methods, the “once-and-for-all” strategy is not applicable. In the asymptotic method, strain energies are derived from systematic sequences of CNBs of the same family, and no reusable reference structures inherently exist. In StrainViz, each molecule’s strain energy is obtained from the complete geometry optimization trajectories of the corresponding fragments, which must be performed anew for every target molecule starting from each fragment’s distorted geometry in the target. Consequently, fragment reuse is not applicable in StrainViz either. The comparative cost advantage of the autodesmotic approach therefore lies in its ability to reuse a pre-built virtual chemical space.

We also agree that, when extending the method to chemically distinct families such as heteroatom-doped or functionalized CNBs, new reference calculations would be required to rebuild or augment the model. This is both logical and natural from the perspective of machine-learning methodology: expanding the model’s chemical coverage demands inclusion of representative new motifs. While the defining equations used in the current work would require modifications for such systems, an even more general alternative is to train the virtual chemical space directly using advanced data-driven machine-learning approaches, without relying on explicit analytical forms. This extension, though beyond the scope of the present study, is a promising direction for future development.

Finally, we recognize that, like all practical models, the autodesmotic framework is not free from limitations and will benefit from future refinements. Nonetheless, it establishes a systematic,

physically interpretable, and generalizable foundation for strain energy evaluation in a broad variety of π -conjugated nanocarbons. The present study thus represents a proof-of-concept demonstration of a generalizable framework that not only enables effective strain-energy predictions but also opens a path toward generic, data-driven methodologies for strain-energy evaluation in extended π -conjugated systems.

Following the above clarifications and discussion, we have made the following changes in the revised manuscript:

(1) We have added the following text at the end of the third paragraph in the “Discussion” section (Pages 21–22):

“We note that homodesmotic-like approaches may also achieve computational savings when pre-computed reference compounds are reused across a series of structurally related target molecules. However, to our knowledge, no rigorous and systematic framework exists that enables such reuse across diverse π -conjugated nanocarbons while maintaining consistency in reference selection. The autodesmotic framework provides such a systematic and generalizable route, formalizing procedures that are otherwise applied in an ad hoc manner. Building this virtual reference space requires a one-time computational investment, analogous to model training in machine learning, after which each new molecule in the same chemical family requires only a single DFT calculation for its SE evaluation.”

(2) We have inserted the following sentence in the fourth paragraph in the “Discussion” section (Page 22):

“Like all practical models, the autodesmotic framework is not free from limitations and will benefit from continued refinement.”

(3) We have added the following text in the “Discussion” section (Page 22):

“The current autodesmotic method has several limitations. First, its implementation is restricted to fully conjugated hydrocarbons and carbon allotropes containing only sp^2 carbons. Extending the framework to heteroatom-doped or functionalized systems will require redefining the reference chemical space to include representative heteroatomic and mixed-hybridization motifs, or alternatively, training fully data-driven models that learn directly from diverse molecular datasets.”

(4) We have revised the wording of “unique” and “uniquely” throughout the manuscript, as detailed below:

Abstract:

*“Here we introduce autodesmotic reactions, a general and efficient framework that maps any strained π -conjugated nanocarbon onto ~~a uniquely~~ **an operationally** defined single-molecule reference while preserving molecular topology and ensuring proper π -energy balance.”*

Page 6:

*“This hypothetical reference is ~~uniquely~~ **systematically** defined within a virtual strain-free chemical space, constructed by physically motivated model fitting to a large training set of planar benzenoid PAHs.”*

Page 20:

*“This single-molecule reference is ~~uniquely~~ located within a virtual strain-free chemical space, constructed from chemically intuitive models trained on a large set of planar benzenoid PAHs. By preserving molecular topology, autodesmotic reactions not only guarantee a ~~unique~~ **well-defined** reference but also ensure proper π -energy balance, addressing two critical factors often overlooked in SE analysis of conjugated systems.”*